# CycleNet: Rethinking Cycle Consistency in Text-Guided Diffusion for Image Manipulation

**Sihan Xu**[1*]   **Ziqiao Ma**[1*]   **Yidong Huang**[1]   **Honglak Lee**[1,2]   **Joyce Chai**[1]

[1]University of Michigan, [2]LG AI Research

{sihanxu,marstin,owenhji,honglak,chaijy}@umich.edu

## Abstract

Diffusion models (DMs) have enabled breakthroughs in image synthesis tasks but lack an intuitive interface for consistent image-to-image (I2I) translation. Various methods have been explored to address this issue, including mask-based methods, attention-based methods, and image-conditioning. However, it remains a critical challenge to enable unpaired I2I translation with pre-trained DMs while maintaining satisfying consistency. This paper introduces CycleNet, a novel but simple method that incorporates cycle consistency into DMs to regularize image manipulation. We validate CycleNet on unpaired I2I tasks of different granularities. Besides the scene and object level translation, we additionally contribute a multi-domain I2I translation dataset to study the physical state changes of objects. Our empirical studies show that CycleNet is superior in translation consistency and quality, and can generate high-quality images for out-of-domain distributions with a simple change of the textual prompt. CycleNet is a practical framework, which is robust even with very limited training data (around 2k) and requires minimal computational resources (1 GPU) to train.

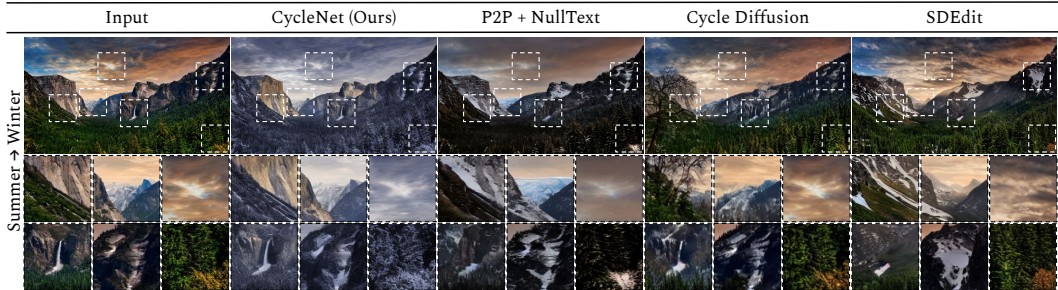

Figure 1: A high-resolution example of CycleNet for diffusion-based image-to-image translation compared to other diffusion-based methods. CycleNet produces high-quality translations with satisfactory consistency. The areas in the boxes are enlarged for detailed comparisons.

## 1   Introduction

Recently, pre-trained diffusion models (DMs) [38, 37, 39] have enabled an unprecedented breakthrough in image synthesis tasks. Compared to GANs [9] and VAEs [22], DMs exhibit superior stability and quality in image generation, as well as the capability to scale up to open-world multimodal data. As such, pre-trained DMs have been applied to image-to-image (I2I) translation, which is to acquire a mapping between images from two distinct domains, e.g., different scenes, different

---

[*]Equal contribution.

37th Conference on Neural Information Processing Systems (NeurIPS 2023).

objects, and different object states. For such translations, text-guided diffusion models typically require mask layers [32, 2, 7, 1] or attention control [10, 30, 25, 35]. However, the quality of masks and attention maps can be unpredictable in complex scenes, leading to semantic and structural changes that are undesirable. Recently, researchers have explored using additional image-conditioning to perform paired I2I translations with the help of a side network [51] or an adapter [31]. Still, it remains an open challenge to adapt pre-trained DMs in *unpaired* I2I translation with a *consistency* guarantee.

We emphasize that *consistency*, a desirable property in image manipulation, is particularly important in unpaired I2I scenarios where there is no guaranteed correspondence between images in the source and target domains. Various applications of DMs, including video prediction and infilling [14], imagination-augmented language understanding [49], robotic manipulation [18, 8] and world models [46], would rely on strong consistency across the source and generated images.

To enable unpaired I2I translation using pre-trained DMs with satisfactory consistency, this paper introduces CycleNet, which allows DMs to translate a source image by conditioning on the input image and text prompts. More specifically, we adopt ControlNet [51] with pre-trained Stable Diffusion (SD) [38] as the latent DM backbone. Motivated by cycle consistency in GAN-based methods [55], CycleNet leverages consistency regularization over the image translation cycle. As illustrated in Figure 2, the image translation cycle includes a forward translation from $x_0$ to $\bar{y}_0$ and a backward translation to $\bar{x}_0$. The key idea of our method is to ensure that when conditioned on an image $c_{\text{img}}$ that falls into the target domain specified by $c_{\text{text}}$, the DM should be able to reproduce this image condition through the reverse process.

We validate CycleNet on I2I translation tasks of different granularities. Besides the scene and object level tasks introduced by Zhu et al. [55], we additionally contribute `ManiCups`, a multi-domain I2I translation dataset for manipulating physical state changes of objects. `ManiCups` contains 6k images of empty cups and cups of coffee, juice, milk, and water, collected from human-annotated bounding boxes. The empirical results demonstrate that compared to previous approaches, CycleNet is superior in translation faithfulness, cycle consistency, and image quality. Our approach is also computationally friendly, which is robust even with very limited training data (around 2k) and requires minimal computational resources (1 GPU) to train. Further analysis shows that CycleNet is a robust zero-shot I2I translator, which can generate faithful and high-quality images for out-of-domain distributions with a simple change of the textual prompt. This opens up possibilities to develop consistent diffusion-based image manipulation models with image conditioning and free-form language instructions.

## 2 Preliminaries

We start by introducing a set of notations to characterize image-to-image translation with DMs.

**Diffusion Models** Diffusion models progressively add Gaussian noise to a source image $z_0 \sim q(z_0)$ through a forward diffusion process and subsequently reverse the process to restore the original image. Given a variance schedule $\beta_1, \ldots, \beta_T$, the forward process is constrained to a Markov chain $q(z_t|z_{t-1}) := \mathcal{N}(z_t; \sqrt{1 - \beta_t}z_{t-1}, \beta_t\mathbf{I})$, in which $z_{1:T}$ are latent variables with dimensions matching $z_0$. The reverse process $p_\theta(z_{0:T})$ is as well Markovian, with learned Gaussian transitions that begin at $z_T \sim \mathcal{N}(0, \mathbf{I})$. Ho et al. [13] noted that the forward process allows the sampling of $z_t$ at any time step $t$ using a closed-form sampling function (Eq. 1).

$$z_t = S(z_0, \varepsilon, t) := \sqrt{\bar{\alpha}_t}z_0 + \sqrt{1 - \bar{\alpha}_t}\varepsilon, \ \varepsilon \sim \mathcal{N}(0, \mathbf{I}) \text{ and } t \sim [1, T] \tag{1}$$

in which $\alpha_t := 1 - \beta_t$ and $\bar{\alpha}_t := \prod_{s=1}^{t} \alpha_s$. Thus, the reverse process can be carried out with a UNet-based network $\varepsilon_\theta$ that predicts the noise $\varepsilon$. By dropping time-dependent variances, the model can be trained according to the objective in Eq. 2.

$$\min_\theta \mathbb{E}_{z_0, \varepsilon, t} \, ||\varepsilon - \varepsilon_\theta(z_t, t)||_2^2 \tag{2}$$

Eq. 2 implies that in principle, one could estimate the original source image $z_0$ given a noised latent $z_t$ at any time $t$. The reconstructed $\bar{z}_0$ can be calculated with the generation function:

$$\bar{z}_0 = G(z_t, t) := \left[z_t - \sqrt{1 - \bar{\alpha}_t}\varepsilon_\theta(z_t, t)\right]/\sqrt{\bar{\alpha}_t} \tag{3}$$

For simplicity, we drop the temporal conditioning $t$ in the following paragraphs.

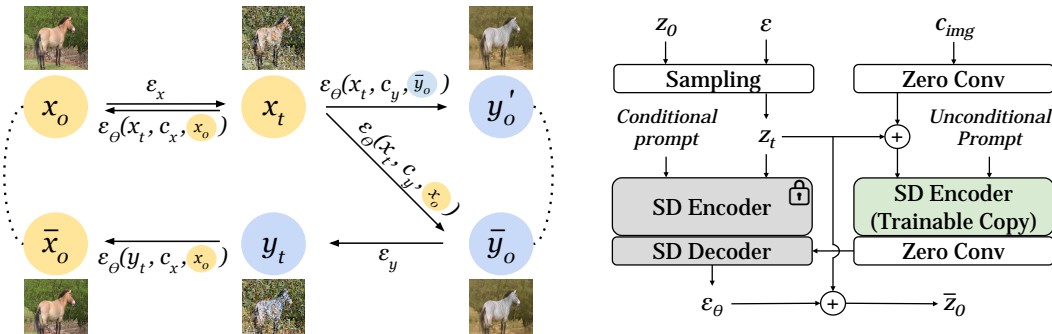

(a) The translation cycle in diffusion-based I2I translation.

(b) CycleNet adopts Stable Diffusion (SD) as the backbone and ControlNet for conditioning.

Figure 2: The image translation cycle includes a forward translation from $x_0$ to $\bar{y}_0$ and a backward translation to $\bar{x}_0$. The key idea of our method is to ensure that when conditioned on an image $c_{\text{img}}$ that falls into the target domain specified by $c_{\text{text}}$, the LDM should reproduce this image condition through the reverse process. The dashed lines indicate the regularization in the loss functions.

**Conditioning in Latent Diffusion Models**    Latent diffusion models (LDMs) like Stable Diffusion [38] can model conditional distributions $p_\theta(z_0|c)$ over condition $c$, e.g., by augmenting the UNet backbone with a condition-specific encoder using cross-attention mechanism [45]. Using textual prompts is the most common approach for enabling conditional image manipulation with LDMs. With a textual prompt $c_z$ as conditioning, LDMs strive to learn a mapping from a latent noised sample $z_t$ to an output image $z_0$, which falls into a domain $\mathcal{Z}$ that is specified by the conditioning prompt. To enable more flexible and robust conditioning in diffusion-based image manipulation, especially a mixture of text and image conditioning, recent work obtained further control over the reverse process with a side network [51] or an adapter [31]. We denote such conditional denoising autoencoder as $\varepsilon_\theta(z_t, c_{\text{text}}, c_{\text{img}})$, where $c_{\text{img}}$ is the image condition and the text condition $c_{\text{text}}$. Eq. 3 can thus be rewritten as:

$$\bar{z}_0 = G(z_t, c_{\text{text}}, c_{\text{img}}) := \left[ z_t - \sqrt{1 - \bar{\alpha}_t}\varepsilon_\theta(z_t, c_{\text{text}}, c_{\text{img}}) \right] / \sqrt{\bar{\alpha}_t} \tag{4}$$

The text condition $c_{\text{text}}$ contains a pair of conditional and unconditional prompts $\{c^+, c^-\}$. A conditional prompt $c^+$ guides the diffusion process towards the images that are associated with it, whereas a negative prompt $c^-$ drives the diffusion process away from those images.

**Consistency Regularization for Unpaired Image-to-Image Translation**    The goal of unpaired image-to-image (I2I) translation is to learn a mapping between two domains $\mathcal{X} \subset \mathbb{R}^d$ and $\mathcal{Y} \subset \mathbb{R}^d$ with unpaired training samples $\{x_i\}$ for $i = 1, \ldots, N$, where $x_i \in \mathcal{X}$ belongs to $X$, and $\{y_j\}$ for $j = 1, \ldots, M$, where $y_j \in \mathcal{Y}$. In traditional GAN-based translation frameworks, the task typically requires two mappings $G : \mathcal{X} \to \mathcal{Y}$ and $F : \mathcal{Y} \to \mathcal{X}$. **Cycle consistency** enforces transitivity between forward and backward translation functions by regularizing pairs of samples, which is crucial in I2I translation, particularly in unpaired settings where no explicit correspondence between images in source and target domains is guaranteed [55, 27, 50, 44]. To ensure cycle consistency, CycleGAN [55] explicitly regularizes the translation cycle, bringing $F(G(x))$ back to the original image $x$, and vice versa for $y$. Motivated by consistency regularization, we seek to enable consistent unpaired I2I translation with LDMs. Without introducing domain-specific generative models, we use one single denoising network $\varepsilon_\theta$ for translation by conditioning it on text and image prompts.

## 3    Method

In the following, we discuss only the translation from domain $\mathcal{X}$ to $\mathcal{Y}$ due to the symmetry of the backward translation. Our goal, at inference time, is to enable LDMs to translate a source image $x_0$ by using it as the image condition $c_{\text{img}} = x_0$, and then denoise the noised latent $y_t$ to $y_{t-1}$ with text prompts $c_{\text{text}} = c_y$. To learn such a translation model $\varepsilon_\theta(y_t, c_y, x_0)$, we consider two types of training objectives. In the following sections, we describe the **cycle consistency regularization** to ensure cycle consistency so that the structures and unrelated semantics are preserved in the generated images, and the **self regularization** to match the distribution of generated images with the target

domain, As illustrated in Figure 2, the image translation cycle includes a forward translation from a source image $x_0$ to $\bar{y}_0$, followed by a backward translation to the reconstructed source image $\bar{x}_0$.

## 3.1 Cycle Consistency Regularization

We assume a likelihood function $P(z_0, c_{\text{text}})$ that the image $z_0$ falls into the data distribution specified by the text condition $c_{\text{text}}$. We consider a generalized case of cycle consistency given the conditioning mechanism in LDMs. If $P(c_{\text{img}}, c_{\text{text}})$ is close to 1, i.e., the image condition $c_{\text{img}}$ falls exactly into the data distribution described by the text condition $c_{\text{text}}$, we should expect that $G(z_t, c_{\text{text}}, c_{\text{img}}) = c_{\text{img}}$ for any noised latent $z_t$. With the translation cycle in Figure 2, the goal is to optimize (1) $\mathcal{L}_{x \to x} = \mathbb{E}_{x_0, \varepsilon_x} ||x_0 - G(x_t, c_x, x_0)||_2^2$; (2) $\mathcal{L}_{y \to y} = \mathbb{E}_{x_0, \varepsilon_x, \varepsilon_y} ||\bar{y}_0 - G(y_t, c_y, \bar{y}_0)||_2^2$; (3) $\mathcal{L}_{x \to y \to x} = \mathbb{E}_{x_0, \varepsilon_x, \varepsilon_y} ||x_0 - G(y_t, c_x, x_0)||_2^2$; and (4) $\mathcal{L}_{x \to y \to y} = \mathbb{E}_{x_0, \varepsilon_x} ||\bar{y}_0 - G(x_t, c_y, \bar{y}_0)||_2^2$.

**Proposition 1** (Cycle Consistency Regularization). *With the translation cycle in Figure 2, a set of consistency losses is given by dropping time-dependent variances:*

$$\mathcal{L}_{x \to x} = \mathbb{E}_{x_0, \varepsilon_x} ||\varepsilon_\theta(x_t, c_x, x_0) - \varepsilon_x||_2^2 \tag{5}$$

$$\mathcal{L}_{y \to y} = \mathbb{E}_{x_0, \varepsilon_x, \varepsilon_y} ||\varepsilon_\theta(y_t, c_y, \bar{y}_0) - \varepsilon_y||_2^2 \tag{6}$$

$$\mathcal{L}_{x \to y \to x} = \mathbb{E}_{x_0, \varepsilon_x, \varepsilon_y} ||\varepsilon_\theta(y_t, c_x, x_0) + \varepsilon_\theta(x_t, c_y, x_0) - \varepsilon_x - \varepsilon_y||_2^2 \tag{7}$$

$$\mathcal{L}_{x \to y \to y} = \mathbb{E}_{x_0, \varepsilon_x} ||\varepsilon_\theta(x_t, c_y, x_0) - \varepsilon_\theta(x_t, c_y, \bar{y}_0)||_2^2 \tag{8}$$

We leave the proof in Section A.2. Proposition 1 states that pixel-level consistency can be acquired by regularizing the conditional denoising autoencoder $\varepsilon_\theta$. Specifically, the **reconstruction loss** $\mathcal{L}_{x \to x}$ and $\mathcal{L}_{y \to y}$ ensures that CycleNet can function as a LDM to reverse an image similar to Eq. 2. The **cycle consistency loss** $\mathcal{L}_{x \to y \to x}$ serves as the transitivity regularization, which ensures that the forward and backward translations can reconstruct the original image $x_0$. The **invariance loss** $\mathcal{L}_{x \to y \to y}$ requires that the target image domain stays invariant under forward translation, i.e., given a forward translation from $x_t$ to $\bar{y}_0$ conditioned on $x_0$, repeating the translation conditioned on $\bar{y}_0$ would reproduce $\bar{y}_0$.

## 3.2 Self Regularization

In the previous section, while $x_0$ is naturally sampled from domain $\mathcal{X}$, we need to ensure that the generated images fall in the target domain $\mathcal{Y}$, i.e., the translation leads to $G(x_t, c_y, x_0) \in \mathcal{Y}$. Our goal is therefore to maximize $P(\bar{y}_0, c_y)$, or equavilently to minimize

$$\mathcal{L}_{\text{LDM}} = -\mathbb{E}_{x_0, \varepsilon_x} P[G(S(x_0, \varepsilon), c_y, x_0), c_y] \tag{9}$$

**Assumption 1** (Domain Smoothness). *For any text condition, $P(\cdot, c_{\text{text}})$ is L-Lipschitz.*

$$\exists L < \infty, \ |P(z_0^1, c_{\text{text}}) - P(z_0^2, c_{\text{text}})| \leq L||z_0^1 - z_0^2||_2 \tag{10}$$

**Proposition 2** (Self Regularization). *Let $\varepsilon_\theta^*$ denote the denoising autoencoder of the pre-trained text-guided LDM backbone. Let $x_t = S(x_0, \varepsilon_x)$ be a noised latent. A self-supervised upper bound of $\mathcal{L}_{\text{LDM}}$ is given by:*

$$\mathcal{L}_{\text{self}} = \mathbb{E}_{x_0, \varepsilon_x} \left[ L \sqrt{\frac{1 - \bar{\alpha}_t}{\bar{\alpha}_t}} ||\varepsilon_\theta(x_t, c_y, x_0) - \varepsilon_\theta^*(x_t, c_y)||_2 \right] + \text{const} \tag{11}$$

Lipschitz assumptions have been widely adopted in diffusion methods [53, 48]. Assumption 1 hypothesizes that similar images share similar domain distributions. A self-supervised upper bound $\mathcal{L}_{\text{self}}$ can be obtained in Proposition 2, which intuitively states that if the output of the conditional translation model does not deviate far from the pre-trained LDM backbone, the outcome image should still fall in the same domain specified by the textual prompt. We leave the proof in Section A.3.

## 3.3 CycleNet

In practice, $\mathcal{L}_{\text{self}}$ can be minimized from the beginning of training by using a ControlNet [51] with pre-trained Stable Diffusion (SD) [38] as the LDM backbone, which is confirmed through preliminary

experiments. As shown in Figure 2, the model keeps the SD encoder frozen and makes a trainable copy in the side network. Additional zero convolution layers are introduced to encode the image condition and control the SD decoder. These zero convolution layers are 1D convolutions whose initial weights and biases vanish and can gradually acquire the optimized parameters from zero. Since the zero convolution layers keep the SD encoder features untouched, $\mathcal{L}_{\text{self}}$ is minimal at the beginning of the training, and the training process is essentially fine-tuning a pre-trained LDM with a side network.

The text condition $c_{\text{text}} = \{c^+, c^-\}$ contains a pair of conditional and unconditional prompts. We keep the conditional prompt in the frozen SD encoder and the unconditional prompt in the ControlNet, so that the LDM backbone focuses on the translation and the side network looks for the semantics that needs modification. For example, to translate an image of summer to winter, we rely on a conditional prompt $l_x =$ "summer" and unconditional prompt $l_y =$ "winter". Specifically, we use CLIP [36] encoder to encode the language prompts $l_x$ and $l_y$ such that $c_x = \{\text{CLIP}(l_x), \text{CLIP}(l_y)\}$ and $c_y = \{\text{CLIP}(l_y), \text{CLIP}(l_x)\}$.

We also note that $\mathcal{L}_{y \to y}$ can be omitted, as $\mathcal{L}_{x \to x}$ can serve the same purpose in the symmetry of the translation cycle from $\mathcal{Y}$ to $\mathcal{X}$, and early experiments confirmed that dropping this term lead to significantly faster convergence. The simplified objective is thus given by:

$$\mathcal{L}_x = \lambda_1 \mathcal{L}_{x \to x} + \lambda_2 \mathcal{L}_{x \to y \to y} + \lambda_3 \mathcal{L}_{x \to y \to x} \tag{12}$$

Consider both translation cycle from $\mathcal{X} \leftrightarrow \mathcal{Y}$, the complete training objective of CycleNet is:

$$\mathcal{L}_{\text{CycleNet}} = \mathcal{L}_x + \mathcal{L}_y \tag{13}$$

The pseudocode for training is given in Algo. 1.

### 3.4 FastCycleNet

Similar to previous cycle-consistent GAN-based models for unpaired I2I translation, there is a trade-off between the image translation quality and cycle consistency. Also, the cycle consistency loss $\mathcal{L}_{x \to y \to x}$ requires deeper gradient descent, and therefore more computation expenses during training (Table 6). In order to speed up the training process in this situation, one may consider further removing $\mathcal{L}_{x \to y \to x}$ from the training objective, and name this variation FastCycleNet. Through experiments, FastCycleNet can achieve satisfying consistency and competitive translation quality, as shown in Table 1. Different variations of models can be chosen depending on the practical needs.

## 4 Experiments

### 4.1 Benchmarks

**Scene/Object-Level Manipulation**   We validate CycleNet on I2I translation tasks of different granularities. We first consider the benchmarks used in CycleGAN by Zhu et al. [55], which contains:

- (Scene Level) Yosemite summer↔winter: We use around 2k images of summer and winter Yosemite, with default prompts "summer" and "winter";
- (Object Level) horse↔zebra: We use around 2.5k images of horses and zebras from the dataset with default prompts "horse" and "zebra";
- (Object Level) apple↔orange: We use around 2k apple and orange images with default prompts of "apple" and "orange".

**State Level Manipulation**   Additionally, we introduce ManiCups[1], a dataset of state-level image manipulation that tasks models to manipulate cups by filling or emptying liquid to/from containers, formulated as a multi-domain I2I translation dataset for object state changes:

- (State Level) ManiCups: We use around 5.7k images of empty cups and cups of coffee, juice, milk, and water for training. The default prompts are set as "empty cup" and "cup of <liquid>". The task is to either empty a full cup or fill an empty cup with liquid as prompted.

---

[1]Our data is available at https://huggingface.co/datasets/sled-umich/ManiCups.

`ManiCups` is curated from human-annotated bounding boxes in publicly available datasets and Bing Image Search (under `Share` license for training and `Modify` for test set). We describe our three-stage data collection pipeline. In the **image collection** stage, we gather raw images of interest from MSCOCO [26], Open Images [23], as well as Bing Image Search API. In the **image extraction** stage, we extract regions of interest from the candidate images and resize them to a standardized size. Specifically, for subsets obtained from MSCOCO and Open Images, we extract the bounding boxes with labels of interest. All bounding boxes with an initial size less than 128×128 are discarded, and the remaining boxes are extended to squares and resized to a standardized size of 512×512 pixels. After this step, we obtained approximately 20k extracted and resized candidate images. We then control the data quality through a **filtering and labeling** stage. Our filtering process first discards replicated images using the L2 distance metric and remove images containing human faces, as well as cups with a front-facing perspective with a CLIP processor. Our labeling process starts with an automatic annotation with a CLIP classifier. To ensure the accuracy of the dataset, three human annotators thoroughly review the collected images, verifying that the images portray a top-down view of a container and assigning the appropriate labels to the respective domains. The resulting `ManiCups` dataset contains 5 domains, including 3 abundant domains (empty, coffee, juice) with more than 1K images in each category and 2 low-resource domains (water, milk) with less than 1K images to facilitate research and analysis in data-efficient learning.

To our knowledge, `ManiCups` is one of the first datasets targeted to the physical state changes of objects, other than stylistic transfers or type changes of objects. The ability to generate consistent state changes based on manipulation is fundamental for future coherent video prediction [14] as well as understanding and planning for physical agents [49, 18, 8, 46]. For additional details on data collection, processing, and statistics, please refer to Appendix B.

## 4.2 Experiment Setup

**Baselines**   We compare our proposed models FastCycleNet and CycleNet to state-of-the-art methods for unpaired or zero-shot image-to-image translation.

- GAN-based methods: CycleGAN [55] and CUT [34];
- Mask-based diffusion methods: Direct inpainting with CLIPSeg [28] and Text2LIVE [2];
- Mask-free diffusion methods: ControlNet with Canny Edge [51], ILVR [5], EGSDE [53], SDEdit [29], Pix2Pix-Zero [35], MasaCtrl [3], CycleDiffusion [48], and Prompt2Prompt [10] with null-text inversion [30].

**Training**   We train our model with a batch size of 4 on only one single A40 GPU.[2] Additional details on the implementations are available in Appendix C.

**Sampling**   As shown in Figure 3, CycleNet has a good efficiency at inference time and more sampling steps lead to better translation quality. We initialize the sampling process with the latent noised input image $z_t$, collected using Equation 1. Following [29], a standard 50-step sampling is applied at inference time with $t = 100$ for fair comparison.

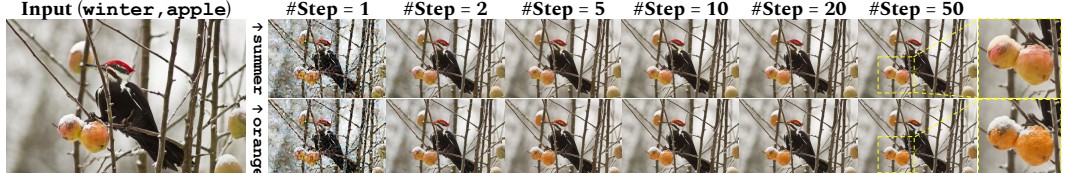

Figure 3: Step skipping during sampling. The source image is from MSCOCO [26].

## 4.3 Qualitative Evaluation

We present qualitative results comparing various image translation models. Due to the space limit, additional examples will be available in Appendix E. In Figure 4, we present the two unpaired translation tasks: `summer`→`winter`, and `horse`→`zebra`. To demonstrate the image quality, translation quality, and consistency compared to the original images, we provide a full image for each test case

---

[2]Our code is available at `https://github.com/sled-group/CycleNet`.

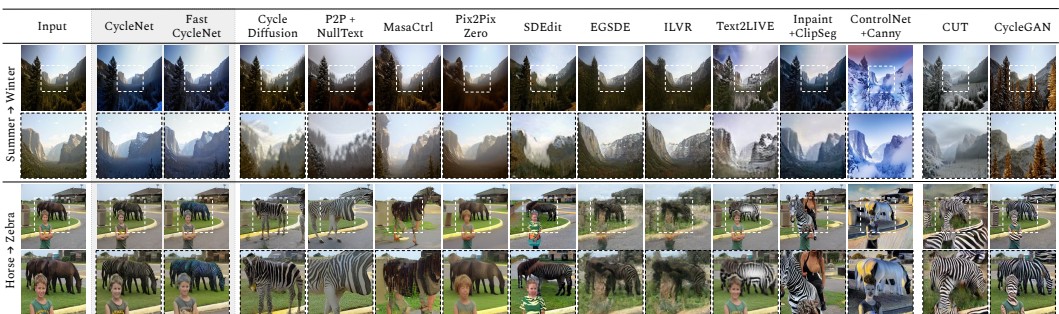

Figure 4: Qualitative comparison in scene/object-level tasks. CycleNet produces high-quality translations with satisfactory consistency. The areas in the boxes are enlarged for detailed comparisons.

| Tasks | summer→winter (Scene level, 256 × 256) | | | | | | | horse→zebra (Object level, 256 × 256) | | | | | | |
|---|---|---|---|---|---|---|---|---|---|---|---|---|---|---|
| Metrics | FID↓ | FID$_{clip}$↓ | CLIP↑ | LPIPS↓ | PSNR↑ | SSIM↑ | L2$^{×10^4}$↓ | FID↓ | FID$_{clip}$↓ | CLIP↑ | LPIPS↓ | PSNR↑ | SSIM↑ | L2$^{×10^4}$↓ |
| GAN-based Methods | | | | | | | | | | | | | | |
| CycleGAN | 133.16 | 18.85 | 22.07 | 0.20 | 16.27 | 0.39 | 3.62 | 77.18 | 27.69 | 28.07 | 0.25 | 18.53 | 0.67 | 1.39 |
| CUT | 180.09 | 23.45 | 24.21 | 0.19 | 20.05 | 0.71 | 1.15 | 45.50 | 21.00 | 29.15 | 0.46 | 13.71 | 0.35 | 2.44 |
| Mask-based Diffusion Methods | | | | | | | | | | | | | | |
| Inpaint + ClipSeg | 246.56 | 79.70 | 21.85 | 0.57 | 12.63 | 0.19 | 2.83 | 187.63 | 40.03 | 26.32 | 0.30 | 15.45 | 0.43 | 2.31 |
| Text2LIVE | 100.63 | 22.59 | 26.03 | 0.22 | 16.51 | 0.67 | 1.74 | 128.21 | 24.46 | 30.51 | 0.14 | 21.05 | 0.81 | 1.03 |
| Mask-free Diffusion Methods | | | | | | | | | | | | | | |
| ControlNet + Canny | 338.24 | 83.26 | 21.77 | 0.59 | 6.05 | 0.09 | 11.30 | 397.71 | 77.68 | 23.88 | 0.61 | 7.37 | 0.07 | 3.89 |
| ILVR | 105.19 | 37.24 | 22.91 | 0.59 | 10.06 | 0.16 | 3.62 | 148.45 | 40.80 | 25.95 | 0.57 | 10.24 | 0.17 | 3.57 |
| EGSDE | 131.00 | 38.74 | 22.96 | 0.44 | 17.68 | 0.27 | 1.53 | 97.61 | 27.79 | 27.31 | 0.41 | 18.05 | 0.29 | 1.44 |
| SDEdit | 330.98 | 79.70 | 21.85 | 0.57 | 12.63 | 0.19 | 2.83 | 398.60 | 83.21 | 24.17 | 0.66 | 9.75 | 0.11 | 4.01 |
| Pix2Pix-Zero | 311.03 | 81.54 | 22.03 | 0.57 | 14.31 | 0.32 | 5.08 | 377.44 | 86.21 | 24.37 | 0.67 | 11.18 | 0.19 | 3.85 |
| MasaCtrl | 106.91 | 52.38 | 20.79 | 0.36 | 16.22 | 0.36 | 3.71 | 333.17 | 68.31 | 21.15 | 0.40 | 16.31 | 0.37 | 1.83 |
| P2P + NullText | 160.00 | 41.12 | 23.31 | 0.37 | 16.84 | 0.39 | 1.73 | 287.45 | 48.93 | 23.91 | 0.36 | 17.20 | 0.41 | 1.68 |
| CycleDiffusion | 243.98 | 62.96 | 22.32 | 0.44 | 15.06 | 0.31 | 2.20 | 347.27 | 66.80 | 25.04 | 0.57 | 11.51 | 0.21 | 3.46 |
| FastCycleNet | **82.48** | 17.61 | **23.62** | 0.14 | **22.45** | **0.57** | 0.91 | **80.75** | **27.23** | 27.36 | 0.32 | 19.29 | 0.51 | 1.31 |
| CycleNet | 82.52 | **17.54** | 23.32 | **0.13** | 22.42 | **0.57** | **0.90** | 81.69 | 28.11 | **28.91** | **0.27** | **20.42** | **0.52** | **1.14** |
| Tasks | empty→coffee (State level, 512 × 512) | | | | | | | coffee→empty (State level, 512 × 512) | | | | | | |
| Metrics | FID↓ | FID$_{clip}$↓ | CLIP↑ | LPIPS↓ | PSNR↑ | SSIM↑ | L2$^{×10^4}$↓ | FID↓ | FID$_{clip}$↓ | CLIP↑ | LPIPS↓ | PSNR↑ | SSIM↑ | L2$^{×10^4}$↓ |
| Mask-based Diffusion Methods | | | | | | | | | | | | | | |
| Inpaint + ClipSeg | 94.14 | 22.96 | 27.12 | 0.29 | 14.1 | 0.65 | 4.81 | 148.11 | 36.18 | 25.95 | 0.33 | 12.82 | 0.57 | 5.52 |
| Text2LIVE | 106.07 | 28.11 | 28.37 | 0.13 | 20.4 | 0.85 | 2.3 | 142.89 | 39.89 | 29.31 | 0.11 | 20.82 | 0.88 | 2.17 |
| Mask-free Diffusion Methods | | | | | | | | | | | | | | |
| SDEdit | **74.08** | 20.61 | **27.75** | 0.38 | 16.82 | 0.61 | 3.32 | 134.87 | 33.38 | 26.04 | 0.15 | 15.83 | 0.67 | 3.48 |
| P2P + NullText | 103.97 | 24.53 | 25.67 | **0.14** | **24.92** | **0.83** | **1.38** | 138.13 | 31.19 | 25.65 | 0.11 | 24.31 | 0.83 | 1.46 |
| CycleDiffusion | 87.39 | 17.59 | 27.39 | 0.18 | 23.36 | 0.81 | 1.67 | 131.25 | 32.52 | 25.73 | **0.10** | **26.47** | **0.85** | **1.13** |
| CycleNet | 105.52 | **16.26** | 27.45 | 0.17 | 21.32 | 0.77 | 1.99 | **95.24** | **28.79** | **27.54** | 0.14 | 21.85 | 0.78 | 1.92 |
| Tasks | empty→juice (State level, 512 × 512) | | | | | | | juice→empty (State level, 512 × 512) | | | | | | |
| Metrics | FID↓ | FID$_{clip}$↓ | CLIP↑ | LPIPS↓ | PSNR↑ | SSIM↑ | L2$^{×10^4}$↓ | FID↓ | FID$_{clip}$↓ | CLIP↑ | LPIPS↓ | PSNR↑ | SSIM↑ | L2$^{×10^4}$↓ |
| Mask-based Diffusion Methods | | | | | | | | | | | | | | |
| Inpaint + ClipSeg | 124.15 | 35.75 | 26.69 | 0.27 | 14.7 | 0.67 | 4.57 | 163.35 | 38.01 | 24.89 | 0.34 | 13.21 | 0.58 | 5.27 |
| Text2LIVE | 116.14 | 31.44 | 29.18 | 0.15 | 16.52 | 0.79 | 3.55 | 157.43 | 45.41 | 26.47 | 0.19 | 18.04 | 0.78 | 3.03 |
| Mask-free Diffusion Methods | | | | | | | | | | | | | | |
| SDEdit | 145.64 | 39.53 | 26.45 | 0.28 | 13.36 | 0.59 | 4.51 | 135.31 | 36.05 | 25.81 | 0.38 | 16.64 | 0.59 | 3.37 |
| P2P + NullText | 148.77 | 37.74 | 26.28 | 0.33 | 17.82 | 0.69 | 2.99 | 149.10 | 36.48 | 23.57 | 0.14 | 22.68 | **0.82** | 1.71 |
| CycleDiffusion | 139.76 | 33.41 | 25.78 | **0.16** | **23.99** | **0.80** | **1.91** | 159.39 | 42.89 | 23.16 | **0.15** | **24.15** | 0.78 | **1.71** |
| CycleNet | **79.02** | **23.42** | **27.75** | 0.17 | 20.18 | 0.76 | 2.27 | **114.33** | **28.79** | **26.17** | 0.17 | 19.78 | 0.74 | 2.37 |

Table 1: A quantitative comparison of various image translation models for the summer→winter, horse→zebra, empty→coffee and juice→empty. Performances are quantified in terms of FID, CLIP Score, LPIPS, PSNR, SSIM, and L2.

and enlarge the boxed areas for detailed comparisons. As presented with the qualitative examples, our methods are able to perform image manipulation with high quality like the other diffusion-based methods, while preserving the structures and unrelated semantics.

In Figure 5, we present qualitative results for filling and emptying a cup: coffee↔empty and empty↔juice. As demonstrated, image editing tasks that require physical state changes pose a significant challenge to baselines, which struggle with the translation itself and/or maintaining strong consistency. CycleNet, again, is able to generate faithful and realistic images that reflect the physical state changes.

## 4.4 Quantitative Evaluation

We further use three types of evaluation metrics respectively to assess the quality of the generated image, the quality of translation, and the consistency of the images. For a detailed explanation of these evaluation metrics, we refer to Appendix C.4.

- **Image Quality**. To evaluate the quality of images, we employ two metrics: The naïve Fréchet Inception Distance (FID) [12] and FID$_{clip}$ [24] with CLIP [36];

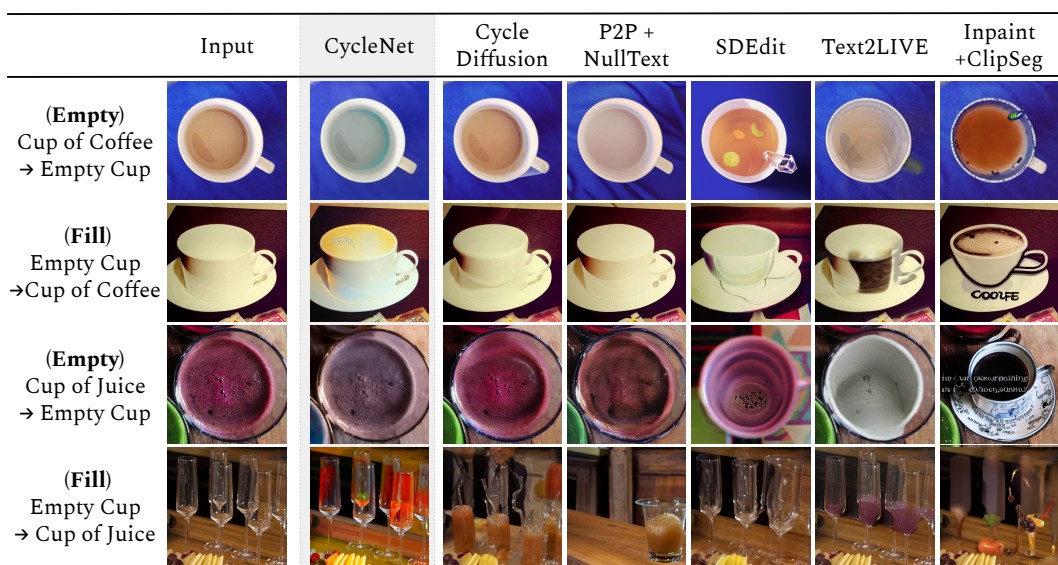

|  | Input | CycleNet | Cycle Diffusion | P2P + NullText | SDEdit | Text2LIVE | Inpaint +ClipSeg |
|---|---|---|---|---|---|---|---|
| (**Empty**) Cup of Coffee → Empty Cup | | | | | | | |
| (**Fill**) Empty Cup →Cup of Coffee | | | | | | | |
| (**Empty**) Cup of Juice → Empty Cup | | | | | | | |
| (**Fill**) Empty Cup → Cup of Juice | | | | | | | |

Figure 5: Qualitative comparison in `coffee`↔`empty` and `empty`↔`juice` tasks.

- **Translation Quality**. We use CLIPScore [11] to quantify the semantic similarity of the generated image and conditional prompt;
- **Translation Consistency**. We measure translation consistency using four different metrics: L2 Distance, Peak Signal-to-Noise Ratio (PSNR) [4], Structural Similarity Index Measure (SSIM) [47], and Learned Perceptual Image Patch Similarity (LPIPS) [52].

The evaluations are performed on the reserved test set. As demonstrated in the quantitative results from Table 1, we observe that some baselines display notably improved consistency in the `ManiCups` tasks. This could possibly be attributed to the fact that a considerable number of the test images were not successfully translated to the target domain, as can be seen in the qualitative results presented in Figure 5. Overall, CycleNet exhibits competitive and comprehensive performance in generating high-quality images in both global and local translation tasks, especially compared to the mask-based diffusion methods. Meanwhile, our methods ensure successful translations that fulfill the domain specified by text prompts while maintaining an impressive consistency from the original images.

## 5    Analysis and Discussions

### 5.1    Ablation study

Recall that FastCycleNet removes the cycle consistency loss $\mathcal{L}_{x \to y \to x}$ from CycleNet. We further remove the invariance loss $\mathcal{L}_{x \to y \to y}$ to understand the role of each loss term. For better control, we initialize the sampling process with the same random noise $\varepsilon$ rather than the latent noised image $z_t$. Table 2 shows our ablation study on `winter`→`summer` task with FID, CLIP score, and LPIPS. When both losses are removed, the model can be considered as a fine-tuned LDM backbone (row 4), which produces a high CLIP similarity score of 24.35. This confirms that the pre-trained LDM backbone can already make qualitative translations, while the LPIPS score of 0.61 implies a poor consistency from the original images. When we introduced the consistency constraint (row 3), the model's LPIPS score improved significantly with a drop of the CLIP score to 19.89. This suggests a trade-off between cycle consistency and translation quality. When we introduced the invariance constraint (row 2), the model achieved the best translation quality with fair consistency. By introducing both constraints (row 1), CycleNet strikes the best balance between consistency at a slight cost of translation quality.

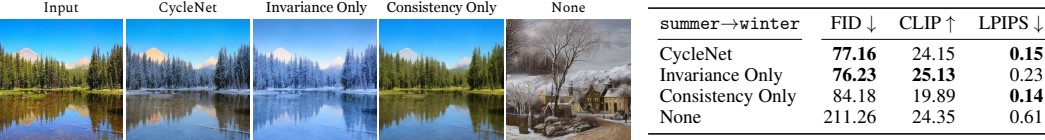

| summer→winter | FID ↓ | CLIP ↑ | LPIPS ↓ |
|---|---|---|---|
| CycleNet | **77.16** | 24.15 | **0.15** |
| Invariance Only | **76.23** | **25.13** | 0.23 |
| Consistency Only | 84.18 | 19.89 | **0.14** |
| None | 211.26 | 24.35 | 0.61 |

Table 2: Ablation study over the cycle consistency loss and invariance loss.

## 5.2 Zero-shot Generalization to Out-of-Distribution Domains

CycleNet performs image manipulation with text and image conditioning, making it potentially generalizable to out-of-distribution (OOD) domains with a simple change of the textual prompt. As illustrated in Figure 6, we demonstrate that CycleNet has a remarkable capability to generate faithful and high-quality images for unseen domains. These results highlight the robustness and adaptability of CycleNet to make the most out of the pre-trained LDM backbone to handle unseen scenarios. This underscores the potential to apply CycleNet for various real-world applications and paves the way for future research in zero-shot learning and OOD generalization.

## 5.3 Translation Diversity

Diversity is an important feature of image translation models. As shown in Figure 6, we demonstrate that CycleNet can generate a variety of images that accurately satisfy the specified translation task in the text prompts, while maintaining consistency.

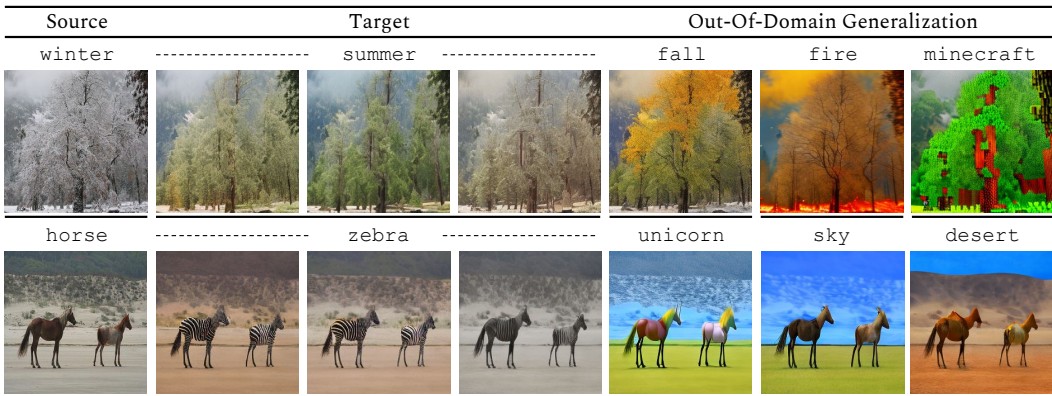

Figure 6: Examples of output diversity and zero-shot generalization to out-of-domain distributions.

## 5.4 Limitations: Trade-off between consistency and translation

There have been concerns that cycle consistency could be too restrictive for some translation task [54]. As shown in Figure 7, while CycleNet maintains a strong consistency over the input image, the quartered apple failed to be translated into its orange equivalence. In GAN-based methods, local discriminators have been proposed to address this issue [56], yet it remains challenging to keep global consistency while making faithful local edits for LDM-based approaches.

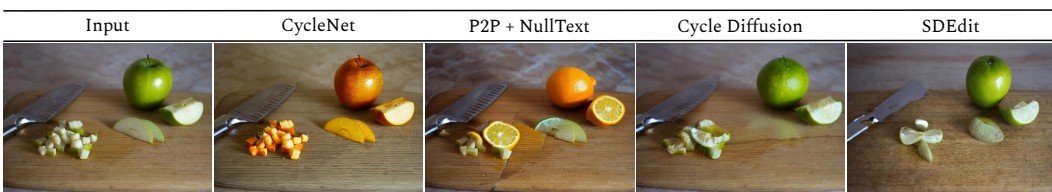

Figure 7: The trade-off between consistency and translation.

## 6 Related Work

### 6.1 Conditional Image Manipulation with Diffusion Models

Building upon Diffusion Probabilistic Models [42, 13], pre-trained Diffusion Models (DMs) [38, 37, 39] have achieved state-of-the-art performance in image generation tasks. Text prompts are the most common protocol to enable conditional image manipulation with DMs, which can be done by fine-tuning a pre-trained DM [19, 20]. Mask-based methods have also been proposed with the help of user-prompted/automatically generated masks [32, 7, 1] or augmentation layers [2]. To refrain from employing additional masks, recent work has explored attention-based alternatives [10, 30, 25, 35].

They first invert the source images to obtain the cross-attention maps and then perform image editing with attention control. The promising performance of these methods is largely dependent on the quality of attention maps, which cannot be guaranteed in images with complicated scenes and object relationships, leading to undesirable changes. Very recently, additional image-conditioning has been explored to perform paired image-to-image translation, using a side network [51] or an adapter [31]. In this work, we follow this line of research and seek to enable unpaired image-to-image translation with pre-trained DMs while maintaining a satisfactory level of consistency.

## 6.2 Unpaired Image-to-Image Translation

Image-to-image translation (I2I) is a fundamental task in computer vision, which is concerned with learning a mapping across images of different domains. Traditional GAN-based methods [16] require instance-level paired data, which are difficult to collect in many domains. To address this limitation, the unpaired I2I setting [55, 27] was introduced to transform an image from the source domain $\mathcal{X}$ into one that belongs to the target domain $\mathcal{Y}$, given only unpaired images from each domain. Several GAN-based methods [55, 50, 27] were proposed to address this problem. In recent years, DPMs have demonstrated their superior ability to synthesize high-quality images, with several applications in I2I translation [40, 5]. With the availability of pre-trained DMs, SDEdit [29] changes the starting point of generation by using a noisy source image that preserves the overall structure. EGSDE [53] combines the merit of ILVR and SDEdit by introducing a pre-trained energy function on both domains to guide the denoising process. While these methods result in leading performance on multiple benchmarks, it remains an open challenge to incorporate pre-trained DMs for high-quality image generation, and at the same time, to ensure translation consistency.

## 6.3 Cycle Consistency in Image Translation

The idea of cycle consistency is to regularize pairs of samples by ensuring transitivity between the forward and backward translation functions [41, 33, 17]. In unpaired I2I translation where explicit correspondence between source and target domain images is not guaranteed, cycle consistency plays a crucial role [55, 21, 27]. Several efforts were made to ensure cycle consistency in diffusion-based I2I translation. UNIT-DDPM [40] made an initial attempt in the unpaired I2I setting, training two DPMs and two translation functions from scratch. Cycle consistency losses are introduced in the translation functions during training to regularize the reverse processes. At inference time, the image generation does not depend on the translation functions, but only on the two DPMs in an iterative manner, leading to sub-optimal performance. Su et al. [43] proposed the DDIB framework that exact cycle consistency is possible assuming zero discretization error, which does not enforce any cycle consistency constraint itself. Cycle Diffusion [48] proposes a zero-shot approach for image translation based on Su et al. [43]'s observation that a certain level of consistency could emerge from DMs, and there is no explicit treatment to encourage cycle consistency. To the best of our knowledge, CycleNet is the first to guarantee cycle consistency in unpaired image-to-image translation using pre-trained diffusion models, with a simple trainable network and competitive performance.

## 7 Conclusion

The paper introduces CycleNet that incorporates the concept of cycle consistency into text-guided latent diffusion models to regularize the image translation tasks. CycleNet is a practical framework for low-resource applications where only limited data and computational power are available. Through extensive experiments on unpaired I2I translation tasks at scene, object, and state levels, our empirical studies show that CycleNet is promising in consistency and quality, and can generate high-quality images for out-of-domain distributions with a simple change of the textual prompt.

**Future Work**    This paper is primarily concerned with the unpaired I2I setting, which utilizes images from unpaired domains during training for domain-specific applications. Although CycleNet demonstrates robust out-of-domain generalization, enabling strong zero-shot I2I translation capabilities is not our focus here. We leave it to our future work to explore diffusion-based image manipulation with image conditioning and free-form language instructions, particularly in zero-shot settings.

## Acknowledgements

This work was supported in part by NSF IIS-1949634, NSF SES-2128623, LG AI Research, and by the Automotive Research Center at the University of Michigan. The authors would like to thank Yinpei Dai and Yuexi Du for their valuable feedback, and extend their special appreciation to Michael de la Paz for granting us permission to use his photograph of summer Yosemite[3] in the teaser.

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

# A  CycleNet Details

## A.1  Algorithmic Details

The pseudocode for training is given as follows.

---
**Algorithm 1:** CycleNet Training Framework

---
1 **Input:** Source image $x_0$, source domain prompt $c_x$, target domain prompt $c_y$, constants $\lambda_{1,2,3}$;
2 **repeat**
3      $x_0 \sim q(x_0)$ ;
4      $t \sim [1, T]$;
5      $\varepsilon_x, \varepsilon_y \sim \mathcal{N}(0, I)$;
6      $x_t = \sqrt{\bar{\alpha}_t} x_0 + \sqrt{1 - \bar{\alpha}_t} \varepsilon_x$;
7      $\varepsilon_\theta^x = \varepsilon_\theta(x_t, c_x, x_0)$;
8      $\varepsilon_\theta^{xy} = \varepsilon_\theta(x_t, c_y, x_0)$;
9      $\bar{y}_0 = (x_t - \sqrt{1 - \bar{\alpha}_t} \varepsilon_\theta^{xy})/\sqrt{\bar{\alpha}_t}$;
10     $\varepsilon_\theta^y = \varepsilon_\theta(x_t, c_y, \bar{y}_0).\text{detach}()$;
11     $y_t = \sqrt{\bar{\alpha}_t} \bar{y}_0 + \sqrt{1 - \bar{\alpha}_t} \varepsilon_y$;
12     $\varepsilon_\theta^{yx} = \varepsilon_\theta(y_t, c_x, x_0)$;
13     Take gradient descent step on
       $\nabla_\theta \left[ \lambda_1 ||\varepsilon_\theta^x - \varepsilon_x||_2^2 + \lambda_2 ||\varepsilon_\theta^{xy} - \varepsilon_\theta^y||_2^2 + \lambda_3 ||\varepsilon_\theta^{xy} + \varepsilon_\theta^{yx} - \varepsilon^x - \varepsilon^y||_2^2 \right]$ ;
14 **until** *converged*;

---

## A.2  Proof of Cycle Consistency Regularization

***x*-Reconstruction Loss**    Dropping the variances, $\mathcal{L}_{x \to x} = \mathbb{E}_{x_0, \varepsilon_x} ||\varepsilon_\theta(x_t, c_x, x_0) - \varepsilon_x||_2^2$.

*Proof.*

$$
\begin{aligned}
\mathcal{L}_{x \to x} &= \mathbb{E}_{x_0, \varepsilon_x} ||x_0 - G(x_t, c_x, x_0)||_2^2 \\
&= \mathbb{E}_{x_0, \varepsilon_x} ||x_0 - [x_t - \sqrt{1 - \bar{\alpha}_t} \varepsilon_\theta(x_t, c_x, x_0)]/\sqrt{\bar{\alpha}_t}||_2^2 \\
&= \mathbb{E}_{x_0, \varepsilon_x} ||x_0 - [\sqrt{\bar{\alpha}_t} x_0 + \sqrt{1 - \bar{\alpha}_t} \varepsilon_x - \sqrt{1 - \bar{\alpha}_t} \varepsilon_\theta(x_t, c_x, x_0)]/\sqrt{\bar{\alpha}_t}||_2^2 \\
&= \mathbb{E}_{x_0, \varepsilon_x} ||\cancel{x_0} - \cancel{x_0} - \sqrt{1 - \bar{\alpha}_t}[\varepsilon_x - \varepsilon_\theta(x_t, c_x, x_0)]/\sqrt{\bar{\alpha}_t}||_2^2 \\
&= \mathbb{E}_{x_0, \varepsilon_x} \sqrt{\frac{1 - \bar{\alpha}_t}{\bar{\alpha}_t}} ||\varepsilon_\theta(x_t, c_x, x_0) - \varepsilon_x||_2^2
\end{aligned}
$$

$\square$

***y*-Reconstruction Loss**    Dropping the variances, $\mathcal{L}_{y \to y} = \mathbb{E}_{x_0, \varepsilon_x, \varepsilon_y} ||\varepsilon_\theta(y_t, c_y, \bar{y}_0) - \varepsilon_y||_2^2$.

*Proof.*

$$
\begin{aligned}
\mathcal{L}_{y \to y} &= \mathbb{E}_{x_0, \varepsilon_x, \varepsilon_y} ||\bar{y}_0 - G(y_t, c_y, \bar{y}_0)||_2^2 \\
&= \mathbb{E}_{x_0, \varepsilon_x, \varepsilon_y} ||\bar{y}_0 - [y_t - \sqrt{1 - \bar{\alpha}_t} \varepsilon_\theta(y_t, c_y, \bar{y}_0)]/\sqrt{\bar{\alpha}_t}||_2^2 \\
&= \mathbb{E}_{x_0, \varepsilon_x, \varepsilon_y} ||\bar{y}_0 - [\sqrt{\bar{\alpha}_t} \bar{y}_0 + \sqrt{1 - \bar{\alpha}_t} \varepsilon_y - \sqrt{1 - \bar{\alpha}_t} \varepsilon_\theta(y_t, c_y, \bar{y}_0)]/\sqrt{\bar{\alpha}_t}||_2^2 \\
&= \mathbb{E}_{x_0, \varepsilon_x, \varepsilon_y} ||\cancel{\bar{y}_0} - \cancel{\bar{y}_0} - \sqrt{1 - \bar{\alpha}_t}[\varepsilon_y - \varepsilon_\theta(y_t, c_y, \bar{y}_0)]/\sqrt{\bar{\alpha}_t}||_2^2 \\
&= \mathbb{E}_{x_0, \varepsilon_x, \varepsilon_y} \sqrt{\frac{1 - \bar{\alpha}_t}{\bar{\alpha}_t}} ||\varepsilon_\theta(y_t, c_y, \bar{y}_0) - \varepsilon_y||_2^2
\end{aligned}
$$

$\square$

**Cycle Consistency Loss** Dropping the variances, $\mathcal{L}_{x\to y\to x} = \mathbb{E}_{x_0,\varepsilon_x,\varepsilon_y} \; ||\varepsilon_\theta(y_t,c_x,x_0) + \varepsilon_\theta(x_t,c_y,x_0) - \varepsilon_x - \varepsilon_y||_2^2$.

*Proof.*

$$
\begin{aligned}
\mathcal{L}_{x\to y\to x} &= \mathbb{E}_{x_0,\varepsilon_x,\varepsilon_y} \; ||x_0 - G(y_t,c_x,x_0)||_2^2 \\
&= \mathbb{E}_{x_0,\varepsilon_x,\varepsilon_y} \; ||x_0 - \left[y_t - \sqrt{1-\bar{\alpha}_t}\varepsilon_\theta(y_t,c_x,x_0)\right]/\sqrt{\bar{\alpha}_t}||_2^2 \\
&= \mathbb{E}_{x_0,\varepsilon_x,\varepsilon_y} \; ||x_0 - \left[\sqrt{\bar{\alpha}_t}\bar{y}_0 + \sqrt{1-\bar{\alpha}_t}\varepsilon_y - \sqrt{1-\bar{\alpha}_t}\varepsilon_\theta(y_t,c_x,x_0)\right]/\sqrt{\bar{\alpha}_t}||_2^2 \\
&= \mathbb{E}_{x_0,\varepsilon_x,\varepsilon_y} \; \left|\left|x_0 - \bar{y}_0 + \sqrt{\frac{1-\bar{\alpha}_t}{\bar{\alpha}_t}}\left[\varepsilon_\theta(y_t,c_x,x_0) - \varepsilon_y\right]\right|\right|_2^2 \\
&= \mathbb{E}_{x_0,\varepsilon_x,\varepsilon_y} \; \left|\left|x_0 - G(x_t,c_y,x_0) + \sqrt{\frac{1-\bar{\alpha}_t}{\bar{\alpha}_t}}\left[\varepsilon_\theta(y_t,c_x,x_0) - \varepsilon_y\right]\right|\right|_2^2 \\
&= \mathbb{E}_{x_0,\varepsilon_x,\varepsilon_y} \; \left|\left|x_0 - x_t/\sqrt{\bar{\alpha}_t} + \sqrt{\frac{1-\bar{\alpha}_t}{\bar{\alpha}_t}}\left[\varepsilon_\theta(x_t,c_y,x_0) + \varepsilon_\theta(y_t,c_x,x_0) - \varepsilon_y\right]\right|\right|_2^2 \\
&= \mathbb{E}_{x_0,\varepsilon_x,\varepsilon_y} \; \left|\left|x_0 - (\sqrt{\bar{\alpha}_t}x_0 + \sqrt{1-\bar{\alpha}_t}\varepsilon_x)/\sqrt{\bar{\alpha}_t} + \sqrt{\frac{1-\bar{\alpha}_t}{\bar{\alpha}_t}}\left[\varepsilon_\theta(x_t,c_y,x_0) + \varepsilon_\theta(y_t,c_x,x_0) - \varepsilon_y\right]\right|\right|_2^2 \\
&= \mathbb{E}_{x_0,\varepsilon_x,\varepsilon_y} \; \left|\left|\cancel{x_0} - \cancel{x_0} + \sqrt{\frac{1-\bar{\alpha}_t}{\bar{\alpha}_t}}\left[\varepsilon_\theta(y_t,c_x,x_0) + \varepsilon_\theta(x_t,c_y,x_0) - \varepsilon_x - \varepsilon_y\right]\right|\right|_2^2 \\
&= \mathbb{E}_{x_0,\varepsilon_x,\varepsilon_y} \; \sqrt{\frac{1-\bar{\alpha}_t}{\bar{\alpha}_t}} \; ||\varepsilon_\theta(y_t,c_x,x_0) + \varepsilon_\theta(x_t,c_y,x_0) - \varepsilon_x - \varepsilon_y||_2^2
\end{aligned}
$$

$\square$

**Invariance Loss** Dropping the variances, $\mathcal{L}_{x\to y\to y} = \mathbb{E}_{x_0,\varepsilon_x} \; ||\varepsilon_\theta(x_t,c_y,x_0) - \varepsilon_\theta(x_t,c_y,\bar{y}_0)||_2^2$.

*Proof.*

$$
\begin{aligned}
\mathcal{L}_{x\to y\to y} &= \mathbb{E}_{x_0,\varepsilon_x} \; ||\bar{y}_0 - G(x_t,c_y,\bar{y}_0)||_2^2 \\
&= \mathbb{E}_{x_0,\varepsilon_x} \; ||G(x_t,c_y,x_0) - G(x_t,c_y,\bar{y}_0)||_2^2 \\
&= \mathbb{E}_{x_0,\varepsilon_x} \; ||\left[\cancel{x_t} - \sqrt{1-\bar{\alpha}_t}\varepsilon_\theta(x_t,c_y,x_0)\right]/\sqrt{\bar{\alpha}_t} - \left[\cancel{x_t} - \sqrt{1-\bar{\alpha}_t}\varepsilon_\theta(x_t,c_y,\bar{y}_0)\right]/\sqrt{\bar{\alpha}_t}||_2^2 \\
&= \mathbb{E}_{x_0,\varepsilon_x} \; \sqrt{\frac{1-\bar{\alpha}_t}{\bar{\alpha}_t}} \; ||\varepsilon_\theta(x_t,c_y,x_0) - \varepsilon_\theta(x_t,c_y,\bar{y}_0)||_2^2
\end{aligned}
$$

$\square$

### A.3 Proof of Self Regularization

Let $\varepsilon_\theta^*$ denote the denoising autoencoder of the pre-trained text-guided LDM backbone with a generation function $G^*$. Note that $x_t = S(x_0,\varepsilon_x)$.

*Proof.*

$$
\begin{aligned}
\mathcal{L}_{\text{LDM}} &= 1 - \mathbb{E}_{x_0,\varepsilon_x} P\left[G(x_t,c_y,x_0),c_y\right] - 1 \\
&= \mathbb{E}_{x_0,\varepsilon_x} \left|1 - P\left[G(x_t,c_y,x_0),c_y\right]\right| - 1 \\
&= \mathbb{E}_{x_0,\varepsilon_x} \left|1 - P\left[G^*(x_t,c_y),c_y\right] + P\left[G^*(x_t,c_y),c_y\right] - P\left[G(x_t,c_y,x_0),c_y\right]\right| - 1 \\
&\leq \mathbb{E}_{x_0,\varepsilon_x} \left|1 - P\left[G^*(x_t,c_y),c_y\right]\right| + \mathbb{E}_{x_0,\varepsilon_x} \left|P\left[G^*(x_t,c_y),c_y\right] - P\left[G(x_t,c_y,x_0),c_y\right]\right| - 1
\end{aligned}
$$

Note that $\mathbb{E}_{x_0,\varepsilon_x}\left|1-P\big[G^*(x_t,c_y),c_y\big]\right| := \mathbb{E}^*_{\text{LDM}}$ is the translation error likelihood of the pre-trained LDM backbone. According to the domain smoothness in Assumption 1, we have:

$$
\begin{aligned}
\mathcal{L}_{\text{LDM}} &\leq \mathbb{E}_{x_0,\varepsilon_x}\left|P\big[G^*(x_t,c_y),c_y\big] - P\big[G(x_t,c_y,x_0),c_y\big]\right| + \mathbb{E}^*_{\text{LDM}} - 1 \\
&\leq \mathbb{E}_{x_0,\varepsilon_x}\, L\left\|G^*(x_t,c_y) - G(x_t,c_y,x_0)\right\| + \mathbb{E}^*_{\text{LDM}} - 1 \\
&= \mathbb{E}_{x_0,\varepsilon_x}\, L\left\|\big[\cancel{x_t} - \sqrt{1-\bar{\alpha}_t}\varepsilon^*_\theta(x_t,c_y)\big]/\sqrt{\bar{\alpha}_t} - \big[\cancel{x_t} - \sqrt{1-\bar{\alpha}_t}\varepsilon_\theta(x_t,c_y,\bar{y}_0)\big]/\sqrt{\bar{\alpha}_t}\right\| + \mathbb{E}^*_{\text{LDM}} - 1 \\
&= \mathbb{E}_{x_0,\varepsilon_x}\, L\sqrt{\frac{1-\bar{\alpha}_t}{\bar{\alpha}_t}}\|\varepsilon_\theta(x_t,c_y,x_0) - \varepsilon^*_\theta(x_t,c_y)\|_2 + \mathbb{E}^*_{\text{LDM}} - 1
\end{aligned}
$$

$\square$

## B   Scientific Artifacts and Licenses

### B.1   The `ManiCups` Dataset

We introduce `ManiCups`, a dataset that tasks image editing models to manipulate cups by filling or emptying liquid to/from containers, curated from human-annotated bounding boxes in publicly available datasets and Bing Image Search (under `Share` license for training and `Modify` for test set). The dataset consists of a total of 5714 images of empty cups and cups of coffee, juice, milk, and water. We describe our three-stage data collection pipeline in the following paragraphs.

In the **Image Collection** stage, we gather raw images of interest from MSCOCO [26], Open Images [23], as well as Bing Image Search API. For the MSCOCO 2017 dataset, we specifically searched for images containing at least one object from the categories of "`bottle`," "`cup`," and "`wine glass`." Regarding the Open Images v7 dataset, our search focused on images with at least one object falling under the categories of "coffee," "`coffee cup`," "`mug`," "`juice`," "`milk`," and "`wine glass`." To conduct the Bing Image Search, we utilized API v7 and employed queries with the formats "empty `<container>`" and "`<container>` of `<liquid>`." The `<container>` category encompasses cups, glasses, and mugs, while the `<liquid>` category includes coffee, juice, water, and milk. We obtained approximately 30,000 image candidates after this step.

During the **Image Extraction** stage, we extract regions of interest from the candidate images and resize them to a standardized size of $512{\times}512$ pixels. Specifically, for subsets obtained from MSCOCO and Open Images, we extract the bounding boxes with labels of interest. To ensure a comprehensive representation, each bounding box is extended by 5% on each side, and the width is adjusted to create a square window. In cases where the square window exceeds the image boundaries, we shift the window inwards. If, due to resizing or shifting, a square window cannot fit within the image, we utilize the `cv2.BORDER_REPLICATE` function to replicate the edge pixels. All bounding boxes with an initial size less than $128{\times}128$ are discarded, and the remaining images are resized to a standardized size of $512{\times}512$ pixels. The same approach is applied to images obtained from the Bing Search API, utilizing `cv2.BORDER_REPLICATE` for edge pixel replication. Following this step, we obtained approximately 20,000 extracted and resized images.

In the **Filtering and Labeling** stage, our focus is on controlling the quality of images and assigning appropriate labels. Our filtering process begins by identifying and excluding replicated images using the L2 distance metric. Subsequently, we leverage the power of CLIP to detect and remove images containing human faces, as well as cups with a front-facing perspective. Additionally, we use CLIP to classify the remaining images into their respective domains. To ensure the accuracy of the dataset, three human annotators thoroughly review the collected images, verifying that the images portray a top-down view of a container and assigning the appropriate labels to the respective domains. The resulting `ManiCups` dataset contains 5 domains, including 3 abundant domains (empty, coffee, juice) with more than 1K images in each category and 2 low-resource domains (water, milk) with less than 1K images to facilitate research and analysis in data-efficient learning.

To our knowledge, `ManiCups` is one of the first datasets targeted to the physical state changes of objects, other than stylistic transfers or type changes of objects. The ability to generate consistent state changes based on manipulation is fundamental for future coherent video prediction [14] as well as understanding and planning for physical agents [49**?** ]. We believe that `ManiCups` is a valuable resource to the community.

## B.2 Dataset Statistics

We present the statistics of the `ManiCups` in Table 3, and other scene/object level datasets in Table 4.

| Domains | Train | Test | Total |
|---------|-------|------|-------|
| Empty Cup | 1256 | 160 | 1416 |
| Cup of Coffee | 1550 | 100 | 1650 |
| Cup of Juice | 1754 | 100 | 1854 |
| Cup of Water | 801 | 50 | 851 |
| Cup of Milk | 353 | 50 | 403 |
| Total | 5714 | 460 | 6174 |

| Domains | Train | Test | Total |
|---------|-------|------|-------|
| Summer | 1231 | 309 | 1540 |
| Winter | 962 | 238 | 1200 |
| Horse | 1067 | 118 | 1194 |
| Zebra | 1334 | 140 | 1474 |
| Apple | 995 | 256 | 1251 |
| Orange | 1019 | 248 | 1267 |

Table 3: The statistics of the `ManiCups` dataset, with 3 abundant domains and 2 low-resource domains.

Table 4: The statistics of the Yosemite summer↔winter, horse↔zebra, and apple↔orange datasets.

## B.3 Licenses of Scientific Artifacts

We present a complete list of references and licenses in Table 5 for all the scientific artifacts we used in this work, including data processing tools, datasets, software source code, and pre-trained weights.

| Data Sources | URL | License |
|--------------|-----|---------|
| FiftyOne (Tool) | Link | Apache v2.0 |
| Summer2Winter Yosemite | Link | ImageNet, See Link |
| Horse2Zebra | Link | ImageNet, See Link |
| Apple2Orange | Link | ImageNet, See Link |
| MSCOCO 2017 | Link | CC BY 4.0 |
| Open Images v7 | Link | Apache v2.0 |
| Bing Image Search API v7 | Link | `Share` (training) & `Modify` (test) |
| **Software Code** | **URL** | **License** |
| Stable Diffusion v1 | Link | CreativeML Open RAIL-M |
| Stable Diffusion v2 | Link | CreativeML Open RAIL++-M |
| ControlNet | Link | Apache v2.0 |
| CUT | Link | Mixed, See Link |
| Prompt2Prompt + NullText | Link | Apache v2.0 |
| Stable Diffusion Inpainting | Link | MIT license |
| Text2LIVE | Link | MIT license |
| Stable Diffusion SDEdit | Link | Apache v2.0 |
| Cycle diffusion | Link | Apache v2.0 |
| ILVR | Link | MIT license |
| EGSDE | Link | N/A |
| P2-weighting | Link | MIT license |
| Pix2pix-zero | Link | MIT license |
| Masactrl | Link | Apache v2.0 |
| **Metric Implementations** | **URL** | **License** |
| FID | Link | Apache v2.0 |
| FCD | Link | MIT license |
| CLIP Score | Link | Apache v2.0 |
| PSNR & SSIM | Link | BSD-3-Clause |
| L2 Distance | Link | BSD-3-Clause |

Table 5: License information for the scientific artifacts used.

# C   Experiment Details

## C.1   Computational Resources

Table 6 illustrates the training performance of CycleNet and FastCycleNet on a single NVIDIA A40 GPU under $256 \times 256$ with batch size of 4. The table provides information on the training speed in seconds per iteration and the memory usage in gigabytes for both models. FastCycleNet exhibits a faster training speed of 1.1 seconds per iteration while consuming 24.5 GB of memory. On the other hand, CycleNet demonstrates a slightly slower training speed of 1.8 seconds per iteration, and it requires 27.9 GB of memory.

| Training | Train Speed (sec/iteration) | Mem Use (GB) |
|---|---|---|
| FastCycleNet | **1.1** | **24.5** |
| CycleNet | 1.8 | 27.9 |

Table 6: Speed of CycleNet and FastCycleNet

## C.2   Hyper-parameter Decisions

We include the major hyper-parameter tuning decisions for reproducibility purposes. In the training of CycleNet, the weights of our three loss functions are respectively set as $\lambda_1 = 1$, $\lambda_2 = 0.1$, and $\lambda_3 = 0.01$. We train the model for 50k steps. Following [29], we initialize the sampling process with the latent noised input image $z_t$, collected using Equation 1. A standard 50-step sampling is applied at inference time with $t = 100$. Our configuration is as follows:

```
model:
  params:
    linear_start: 0.00085
    linear_end: 0.0120
    num_timesteps_cond: 1
    timesteps: 1000
    image_size: 64
    channels: 4
    cond_stage_trainable: false
    monitor: val/loss_simple_ema
    scale_factor: 0.18215
    use_ema: False
    only_mid_control: False
    recon_weight: 1          #lambda1
    disc_weight: 0.1         #lambda2
    cycle_weight: 0.01       #lambda3
    disc_mode: eps
    consis_weight: 0.1
```

For more details, please refer to the supplementary codes.

## C.3   Baseline Implementations

- **CycleGAN**: We used some of the results provided in [34].
- **CUT**: We used the official code[4] provided by the authors.
- **Inpainting + ClipSeg**: we modified from the gradio[5] provided by the community.
- **Text2LIVE**: We used the official code[6] provided by the authors.
- **ILVR**: We first pre-train the diffusion model using P2-weighting [6], and then generated output using the official code[7] provided by the authors.

---

[4]https://github.com/taesungp/contrastive-unpaired-translation
[5]https://huggingface.co/spaces/multimodalart/stable-diffusion-inpainting
[6]https://github.com/omerbt/Text2LIVE
[7]https://github.com/jychoi118/ilvr_adm

- **EGSDE**: We first pre-train the diffusion model using P2-weighting [6], and then generated output using the official code[8] provided by the authors.
- **SDEdit**: We used the community implementation[9] of SDEdit based on stable diffusion.
- **CycleDiffusion**: We modified from the official gradio[10] provided by the authors.
- **Prompt2Prompt + NullText**: We used the official code[11] provided by the authors.
- **Masactrl**: We used the official code[12] provided by the authors.
- **Pix2pix-zero**: We used the official code[13] provided by the authors, and we generated the summer2winter direction assets following the scripts provided by the authors.

### C.4  Evaluation Metrics Explained

**Image Quality**    To evaluate the quality of images, we employ two metrics.

- **Fréchet Inception Distance (FID) [12]** is a widely used metric in image generation tasks. A lower FID score indicates better image quality and more realistic samples.
- **FID$_{\text{clip}}$ [24]** combines the FID metric with features extracted with a CLIP [36] encoder, providing better assessment of image quality. Similar to FID, a lower FID$_{\text{clip}}$ score represents better image quality.

**Translation Quality**    To measure to what extent is the translation successful, we use the **CLIP Score** [11], i.e., the CLIP model [36] to obtain the latent representations of images and prompts, and then calculate the cosine similarity between them. A higher CLIP score indicates a stronger similarity between the generated image and the text prompt, thus better translation.

**Translation Consistency**    We measure translation consistency using four different metrics.

- **L2 Distance** is a measure of the Euclidean distance between two images. A lower L2 distance indicates higher similarity and better translation consistency.
- **Peak Signal-to-Noise Ratio (PSNR) [4]** measures the ratio between the maximum possible power of a signal and the power of corrupting noise. A higher PSNR score indicates better translation consistency.
- **Structural Similarity Index Measure (SSIM) [47]** is a metric used to compare the structural similarity between two images. A higher SSIM score suggests higher similarity and better translation consistency.
- **Learned Perceptual Image Patch Similarity (LPIPS) [52]** is a comprehensive evaluation metric for the perceptual similarity between two images. A lower LPIPS score indicates higher perceptual similarity.

## D    Broader Impact

While CycleNet holds great promise, it is essential to address potential broader impacts, including ethical, legal, and societal considerations. One significant concern is copyright infringement. As an image translation model, CycleNet can potentially be used to create derived works from artists' original images, raising the potential for copyright violations. To safeguard the rights of content creators and uphold the integrity of the creative economy, it is imperative to prioritize careful measures and diligently adhere to licensing requirements. Another critical aspect to consider is the potential for fabricated images to contribute to deception and security threats. If misused or accessed by malicious actors, the ability to generate realistic fake images could facilitate misinformation campaigns, fraud, and even identity theft. This underscores the need for responsible deployment and robust security measures to mitigate such risks. CycleNet leverages pre-trained latent diffusion models, which may encode biases that lead to fairness issues. It is worth noting that the proposed method is currently

---

[8]https://github.com/ML-GSAI/EGSDE

[9]https://huggingface.co/docs/diffusers

[10]https://huggingface.co/spaces/ChenWu98/Stable-CycleDiffusion

[11]https://github.com/google/prompt-to-prompt

[12]https://github.com/TencentARC/MasaCtrl

[13]https://github.com/pix2pixzero/pix2pix-zero

purely algorithmic, devoid of pre-training on web-scale datasets itself. By acknowledging and actively addressing these broader impacts, we can work towards harnessing the potential of CycleNet while prioritizing ethics, legality, and societal well-being.

# E  Addendum to Results

## E.1  Choice of Pre-trained LDM Backbone

For all experiments in the main paper, we use Stable Diffusion 2.1[14] as the pre-trained LDM backbone. We additionally attach a quantitative comparison of CycleNet using Stable Diffusion 1.5,[15] which indicates marginal differences in practice.

| Tasks | summer→winter ($512 \times 512$) | | | | | | | horse→zebra ($512 \times 512$) | | | | | | |
|---|---|---|---|---|---|---|---|---|---|---|---|---|---|---|
| Metrics | FID↓ | FID$_{clip}$↓ | CLIP↑ | LPIPS↓ | PSNR↑ | SSIM↑ | L2$^{\times 10^5}$↓ | FID↓ | FID$_{clip}$↓ | CLIP↑ | LPIPS↓ | PSNR↑ | SSIM↑ | L2$^{\times 10^5}$↓ |
| SD v1.5 | 83.45 | **15.21** | 23.56 | 0.17 | 25.40 | **0.74** | 1.39 | **73.34** | 25.42 | **27.77** | 0.25 | 20.96 | 0.64 | 0.79 |
| SD v2.1 | **79.79** | 15.39 | **24.12** | **0.15** | **25.88** | 0.69 | **1.23** | 76.83 | **24.78** | 25.27 | **0.08** | **26.21** | **0.74** | **0.59** |

Table 7: A quantitative comparison using Stable Diffusion 1.5 and 2.1.

## E.2  Image Resolution

We notice that the Stable Diffusion [38] backbone is pre-trained in multiple stages, initially at a resolution of $256 \times 256$ and followed by another stage at a resolution of $512 \times 512$ or beyond. This could potentially lead to the under-performance of zero-shot diffusion-based methods on summer→winter and horse→zebra, which are at a resolution of $256 \times 256$. We repeat the experiment on these two tasks at a generation resolution of $512 \times 512$ and report the results in Table 8. It's important to acknowledge that this particular setting presents an unfair comparison with considerable challenges for our methods, primarily because the training images are set at a resolution of $256 \times 256$, yet our model is expected to adapt to a higher resolution. Still, we observe a competitive performance of our models, especially in summer→winter. The diminished effectiveness of our method in the horse→zebra task can be attributed to the fact that the zebra patterns, initially acquired at a resolution of $256 \times 256$, lose realism and become overly dense when scaled up to $512 \times 512$ (see Example 3, Figure 11). This limitation can potentially be addressed by scaling to images with multiple resolutions.

| Tasks | summer→winter (Global, $512 \times 512$) | | | | | | | horse→zebra (Local, $512 \times 512$) | | | | | | |
|---|---|---|---|---|---|---|---|---|---|---|---|---|---|---|
| Metrics | FID↓ | FID$_{clip}$↓ | CLIP↑ | LPIPS↓ | PSNR↑ | SSIM↑ | L2$^{\times 10^4}$↓ | FID↓ | FID$_{clip}$↓ | CLIP↑ | LPIPS↓ | PSNR↑ | SSIM↑ | L2$^{\times 10^4}$↓ |
| *Mask-based Diffusion Methods* | | | | | | | | | | | | | | |
| Inpaint + ClipSeg | 168.43 | 46.48 | 26.57 | 0.65 | 9.01 | 0.13 | 4.07 | 79.14 | 26.05 | 28.71 | 0.29 | 16.89 | 0.60 | 1.95 |
| Text2LIVE | 86.12 | 18.30 | 25.98 | 0.27 | 16.83 | 0.68 | 1.67 | 103.14 | 22.71 | 31.55 | 0.16 | 20.98 | 0.81 | 2.08 |
| *Mask-free Diffusion Methods* | | | | | | | | | | | | | | |
| ControlNet + Canny | 179.17 | 44.71 | 25.01 | 0.61 | 9.97 | 0.19 | 7.23 | 112.63 | 68.31 | 27.22 | 0.6 | 8.52 | 0.06 | 8.65 |
| ILVR | 101.26 | 27.72 | 21.71 | 0.58 | 9.87 | 0.18 | 3.70 | 194.92 | 45.44 | 23.90 | 056 | 10.06 | 0.24 | 7.27 |
| EGSDE | 108.71 | 35.89 | 21.80 | 0.37 | 19.84 | 0.39 | 2.40 | 161.26 | 39.94 | 25.52 | 0.36 | 20.67 | 0.40 | 2.15 |
| SDEdit | 90.51 | 21.23 | 23.26 | 0.30 | 18.59 | 0.43 | 1.39 | 63.04 | 22.65 | 27.97 | 0.33 | 18.49 | 0.44 | 2.96 |
| Pix2Pix-Zero | 88.79 | 83.78 | 23.63 | 0.29 | 20.91 | 0.62 | 2.15 | 115.52 | 29.74 | 27.42 | 0.37 | 18.18 | 0.57 | 3.14 |
| MasaCtrl | 114.83 | 29.18 | 17.11 | 0.37 | 14.66 | 0.43 | 2.28 | 239.61 | 47.48 | 21.15 | 0.41 | 16.31 | 0.37 | 1.83 |
| P2P + NullText | 92.65 | 22.46 | **24.82** | 0.24 | 20.19 | 0.66 | 1.15 | 106.83 | 26.49 | 26.57 | 0.21 | 21.45 | 0.66 | 2.04 |
| CycleDiffusion | 84.52 | 20.85 | 24.40 | 0.24 | 21.66 | 0.68 | **0.98** | 41.17 | 18.10 | 29.09 | 0.29 | 19.41 | 0.61 | 2.53 |
| FastCycleNet | **78.43** | **14.99** | 24.33 | 0.16 | 25.81 | **0.76** | 1.24 | 72.68 | 25.34 | 24.42 | 0.13 | **26.74** | 0.72 | 1.12 |
| CycleNet | 79.79 | 15.39 | 24.12 | **0.15** | **25.88** | 0.69 | 1.23 | 76.83 | 24.78 | 25.27 | **0.08** | 26.21 | **0.74** | **0.59** |

Table 8: A quantitative comparison of various image translation models for the summer→winter and horse→zebra at $512 \times 512$.

## E.3  Sudden Convergence

As shown in Figure 8, CycleNet can translate an input image of summer to winter at the beginning of training with no consistency observed. Similar to ControlNet [51], CycleNet also demonstrates the sudden convergence phenomenon, which usually happens around 3k to 8k iterations of training.

## E.4  Addtional Quantitative Results

In Table 9, we present the complete numerical performance of the state-changing tasks on ManiCups. In general, we found that emptying a cup is a more challenging image editing task for most of

---

[14]https://huggingface.co/stabilityai/stable-diffusion-2-1
[15]https://huggingface.co/runwayml/stable-diffusion-v1-5

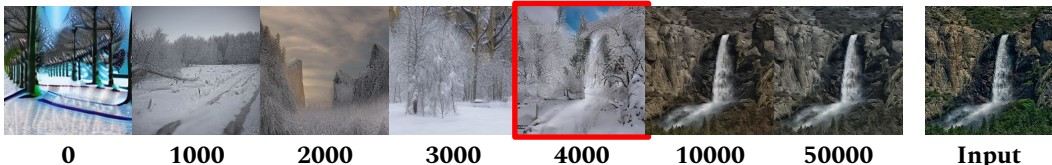

| 0 | 1000 | 2000 | 3000 | 4000 | 10000 | 50000 | Input |

Figure 8: Sudden convergence appears at around 3k - 8k in our experiments.

the existing computation models, compared to filling a cup with liquid. CycleNet shows superior performance, especially in domains with abundant training samples (coffee and juice). We also observe that the performance is marginally less competitive in low-resource domains (milk and water). We leave it to future work to explore more data-efficient models that fine-tune pre-trained latent diffusion models with cycle consistency granted.

| Tasks | empty→coffee (512 × 512) | | | | | | | coffee→empty (512 × 512) | | | | | | |
|---|---|---|---|---|---|---|---|---|---|---|---|---|---|---|
| Metrics | FID↓ | FID$_{clip}$↓ | CLIP↑ | LPIPS↓ | PSNR↑ | SSIM↑ | L2$^{×10^4}$↓ | FID↓ | FID$_{clip}$↓ | CLIP↑ | LPIPS↓ | PSNR↑ | SSIM↑ | L2$^{×10^4}$↓ |
| Mask-based Diffusion Methods | | | | | | | | | | | | | | |
| Inpaint + ClipSeg | 94.14 | 22.96 | 27.12 | 0.29 | 14.1 | 0.65 | 4.81 | 148.11 | 36.18 | 25.95 | 0.33 | 12.82 | 0.57 | 5.52 |
| Text2LIVE | 106.07 | 28.11 | 28.37 | 0.13 | 20.4 | 0.85 | 2.3 | 142.89 | 39.89 | 29.31 | 0.11 | 20.82 | 0.88 | 2.17 |
| Mask-free Diffusion Methods | | | | | | | | | | | | | | |
| SDEdit | **74.08** | 20.61 | **27.75** | 0.38 | 16.82 | 0.61 | 3.32 | 134.87 | 33.38 | 26.04 | 0.15 | 15.83 | 0.67 | 3.48 |
| P2P + NullText | 103.97 | 24.53 | 25.67 | **0.14** | **24.92** | **0.83** | **1.38** | 138.13 | 31.19 | 25.65 | 0.11 | 24.31 | 0.83 | 1.46 |
| CycleDiffusion | 87.39 | 17.59 | 27.39 | 0.18 | 23.36 | 0.81 | 1.67 | 131.25 | 32.52 | 25.73 | **0.10** | **26.47** | **0.85** | **1.13** |
| CycleNet | 105.52 | **16.26** | 27.45 | 0.17 | 21.32 | 0.77 | 1.99 | **95.24** | **28.79** | **27.54** | 0.14 | 21.85 | 0.78 | 1.92 |
| Tasks | empty→juice (512 × 512) | | | | | | | juice→empty (512 × 512) | | | | | | |
| Metrics | FID↓ | FID$_{clip}$↓ | CLIP↑ | LPIPS↓ | PSNR↑ | SSIM↑ | L2$^{×10^4}$↓ | FID↓ | FID$_{clip}$↓ | CLIP↑ | LPIPS↓ | PSNR↑ | SSIM↑ | L2$^{×10^4}$↓ |
| Mask-based Diffusion Methods | | | | | | | | | | | | | | |
| Inpaint + ClipSeg | 124.15 | 35.75 | 26.69 | 0.27 | 14.7 | 0.67 | 4.57 | 163.35 | 38.01 | 24.89 | 0.34 | 13.21 | 0.58 | 5.27 |
| Text2LIVE | 116.14 | 31.44 | 29.18 | 0.15 | 16.52 | 0.79 | 3.55 | 157.43 | 45.41 | 26.47 | 0.19 | 18.04 | 0.78 | 3.03 |
| Mask-free Diffusion Methods | | | | | | | | | | | | | | |
| SDEdit | 145.64 | 39.53 | 26.45 | 0.28 | 13.36 | 0.59 | 4.51 | 135.31 | 36.05 | 25.81 | 0.38 | 16.64 | 0.59 | 3.37 |
| P2P + NullText | 148.77 | 37.74 | 26.28 | 0.33 | 17.82 | 0.69 | 2.99 | 149.10 | 36.48 | 23.57 | 0.14 | 22.68 | **0.82** | **1.71** |
| CycleDiffusion | 139.76 | 33.41 | 25.78 | **0.16** | **23.99** | **0.80** | **1.91** | 159.39 | 42.89 | 23.16 | **0.15** | **24.15** | 0.78 | **1.71** |
| CycleNet | **79.02** | 23.42 | **27.75** | 0.17 | 20.18 | 0.76 | 2.27 | **114.33** | **28.79** | 26.17 | 0.17 | 19.78 | 0.74 | 2.37 |
| Tasks | empty→milk (Low Resource, 512 × 512) | | | | | | | milk→empty (Low Resource, 512 × 512) | | | | | | |
| Metrics | FID↓ | FID$_{clip}$↓ | CLIP↑ | LPIPS↓ | PSNR↑ | SSIM↑ | L2$^{×10^4}$↓ | FID↓ | FID$_{clip}$↓ | CLIP↑ | LPIPS↓ | PSNR↑ | SSIM↑ | L2$^{×10^4}$↓ |
| Mask-based Diffusion Methods | | | | | | | | | | | | | | |
| Inpaint + ClipSeg | 138.81 | 30.13 | 27.11 | 0.28 | 14.97 | 0.66 | 4.47 | 185.92 | 41.37 | 26.27 | 0.35 | 13.09 | 0.57 | 5.59 |
| Text2LIVE | 110.73 | 30.69 | 30.75 | 0.13 | 20.11 | 0.85 | 2.37 | 166.72 | 49.97 | 28.12 | 0.17 | 19.15 | 0.83 | 2.65 |
| Mask-free Diffusion Methods | | | | | | | | | | | | | | |
| SDEdit | 125.75 | 28.97 | 28.38 | 0.38 | 16.91 | 0.61 | 3.39 | 142.41 | 39.09 | **26.88** | 0.36 | 16.95 | 0.62 | 3.38 |
| P2P + NullText | 125.54 | 29.57 | 25.51 | **0.13** | **25.18** | **0.84** | **1.34** | 147.65 | 40.16 | 24.99 | **0.12** | **22.45** | **0.83** | **1.76** |
| CycleDiffusion | 132.24 | 27.27 | 26.71 | 0.16 | 24.21 | 0.81 | 1.75 | 151.01 | 38.14 | 25.36 | 0.15 | 24.19 | 0.81 | 1.96 |
| CycleNet | **97.07** | **25.84** | 27.61 | 0.19 | 22.58 | 0.77 | 1.74 | **121.99** | **32.21** | 26.29 | 0.15 | 21.65 | 0.79 | 1.93 |
| Tasks | empty→water (Low Resource, 512 × 512) | | | | | | | water→empty (Low Resource, 512 × 512) | | | | | | |
| Metrics | FID↓ | FID$_{clip}$↓ | CLIP↑ | LPIPS↓ | PSNR↑ | SSIM↑ | L2$^{×10^4}$↓ | FID↓ | FID$_{clip}$↓ | CLIP↑ | LPIPS↓ | PSNR↑ | SSIM↑ | L2$^{×10^4}$↓ |
| Mask-based Diffusion Methods | | | | | | | | | | | | | | |
| Inpaint + ClipSeg | 135.87 | 32.87 | 26.48 | 0.28 | 15.37 | 0.67 | 4.19 | 191.57 | 42.49 | 24.66 | 0.28 | 14.98 | 0.63 | 4.59 |
| Text2LIVE | 133.04 | 42.85 | 30.23 | 0.16 | 21.09 | 0.81 | 2.08 | 172.72 | 50.06 | 26.64 | 0.13 | 21.37 | 0.84 | 2.05 |
| Mask-free Diffusion Methods | | | | | | | | | | | | | | |
| SDEdit | 147.25 | 39.49 | 28.08 | 0.37 | 17.24 | 0.62 | 3.24 | 143.17 | 39.79 | **27.14** | 0.45 | 16.53 | 0.55 | 3.44 |
| P2P + NullText | 132.69 | 30.11 | 25.67 | **0.13** | **25.26** | **0.85** | **1.34** | 171.14 | 38.86 | 23.80 | **0.11** | **25.41** | **0.85** | **1.25** |
| CycleDiffusion | 146.11 | 31.40 | 24.43 | **0.13** | 22.38 | 0.74 | 1.35 | 157.81 | 35.98 | 25.24 | 0.16 | 24.57 | 0.82 | 1.39 |
| CycleNet | **95.97** | **26.25** | **28.38** | 0.16 | 22.11 | 0.79 | 1.84 | **133.24** | **31.11** | 25.75 | 0.16 | 22.69 | 0.77 | 1.70 |

Table 9: The complete quantitative comparison of the state change tasks on `ManiCups`.

### E.5 Additional Qualitative Results

We present additional qualitative examples in Figure 9 and 10.

### E.6 Additional High-resolution Qualitative Results

We present additional high-resolution examples in Figure 11.

### E.7 Additional Out-of-Domain Examples

We present additional OOD examples in Figure 12.

| Empty Cup → | cup of coffee | cup of juice | cup of milk | cup of water |
| --- | --- | --- | --- | --- |

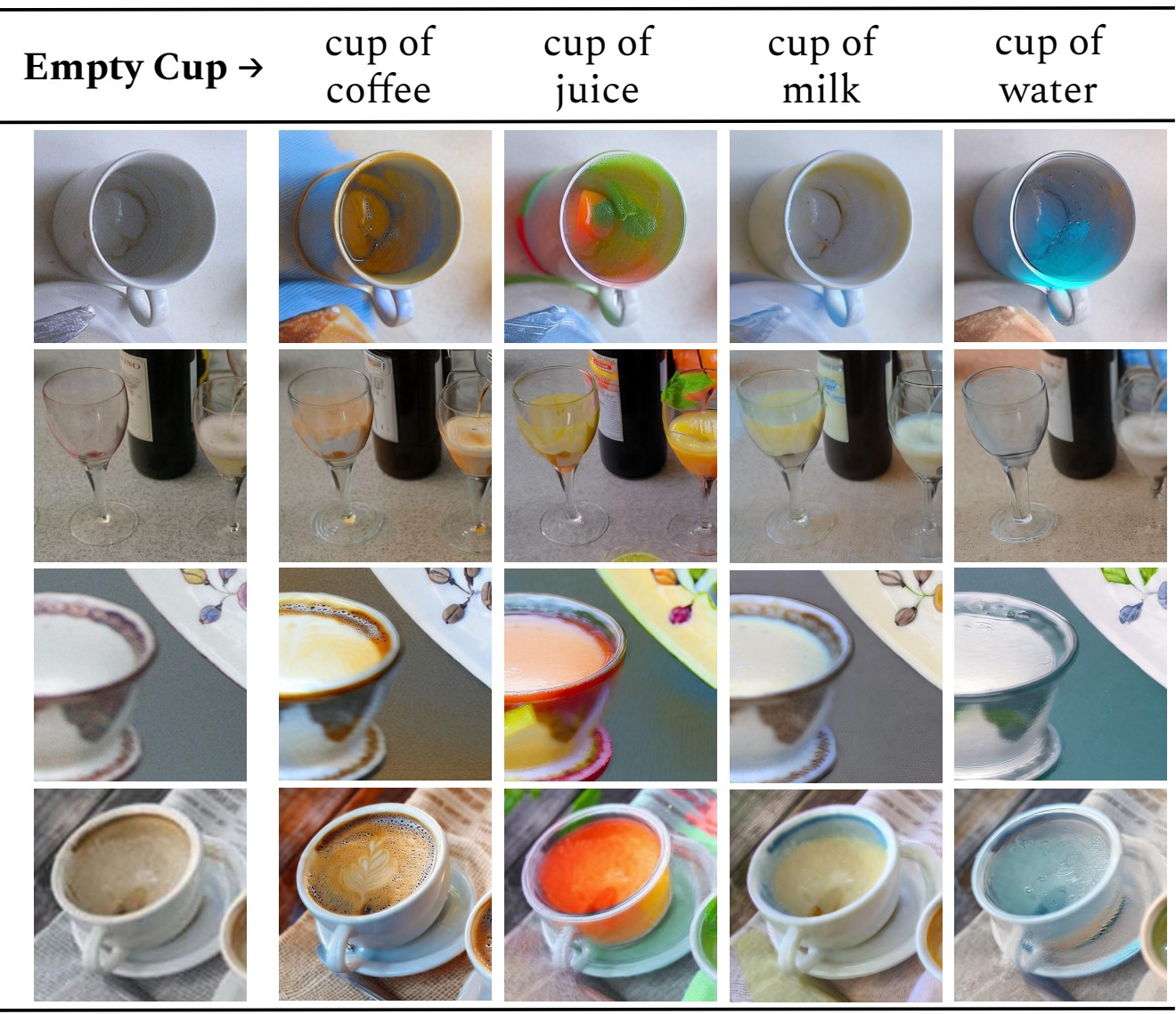

Figure 9: Additional qualitative results using CycleNet on Manicups dataset.

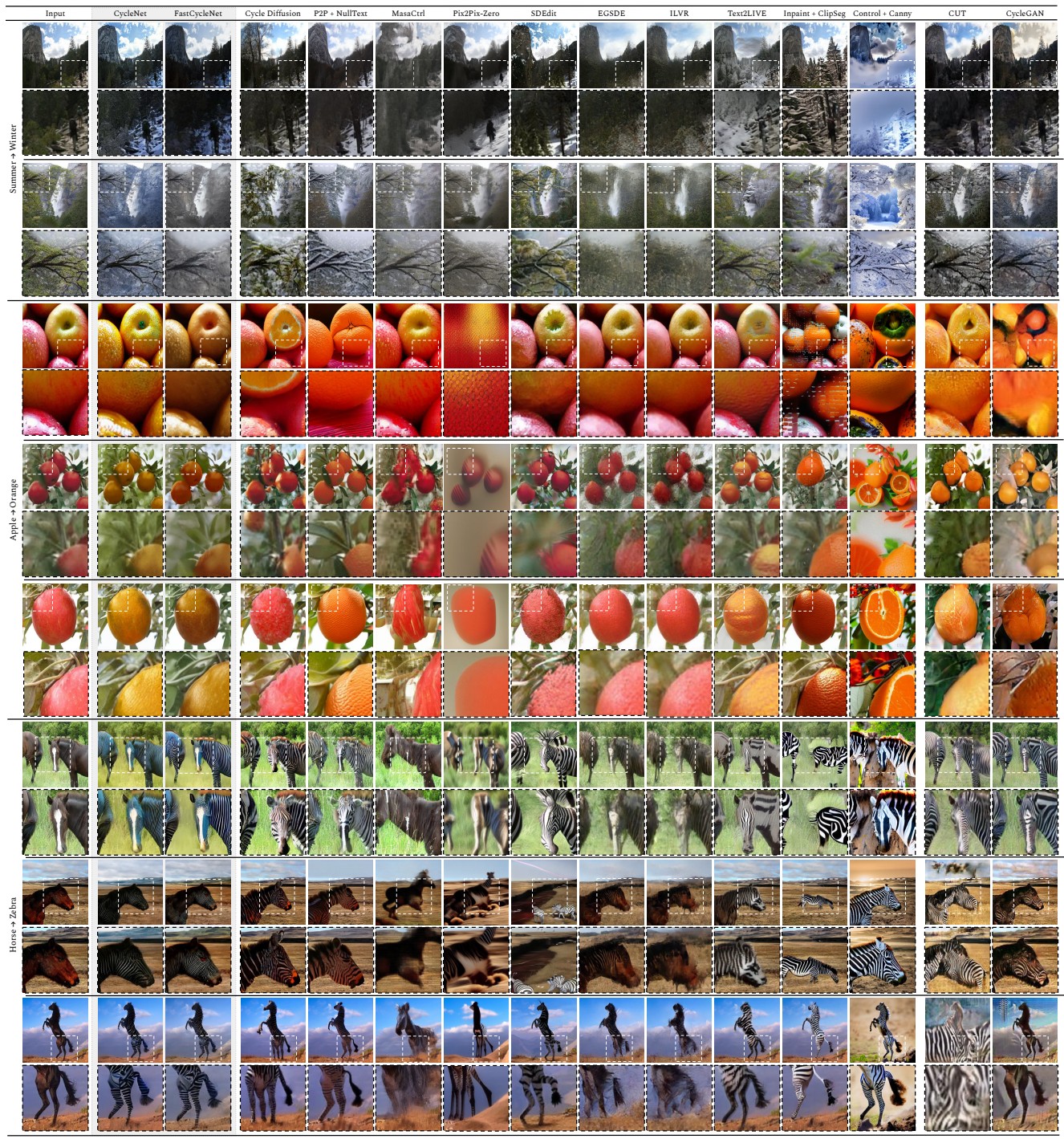

Figure 10: Additional qualitative comparison of our method with other baselines.

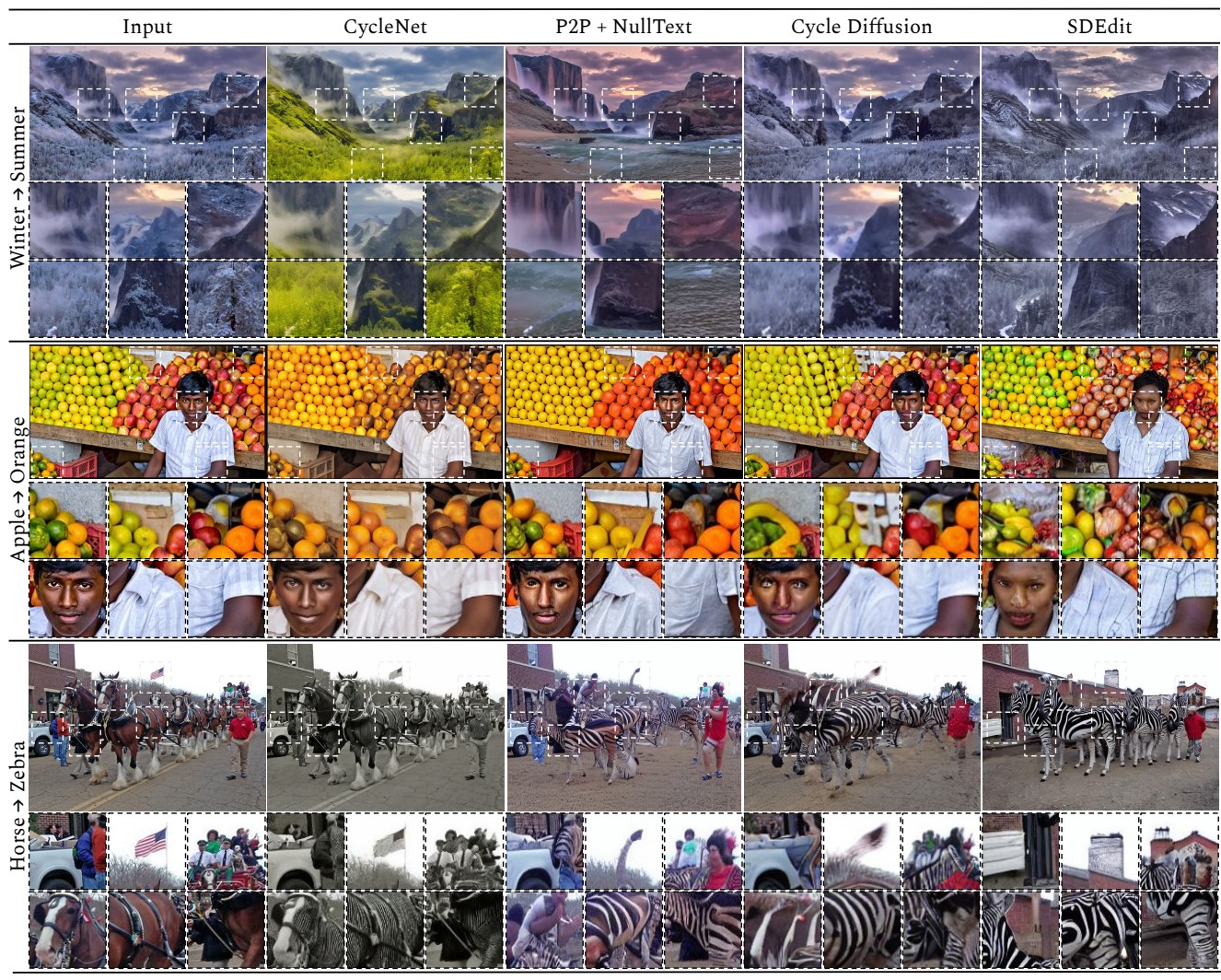

Figure 11: Additional high-resolution examples from Yosemite summer↔winter HD [15] and MSCOCO [26].

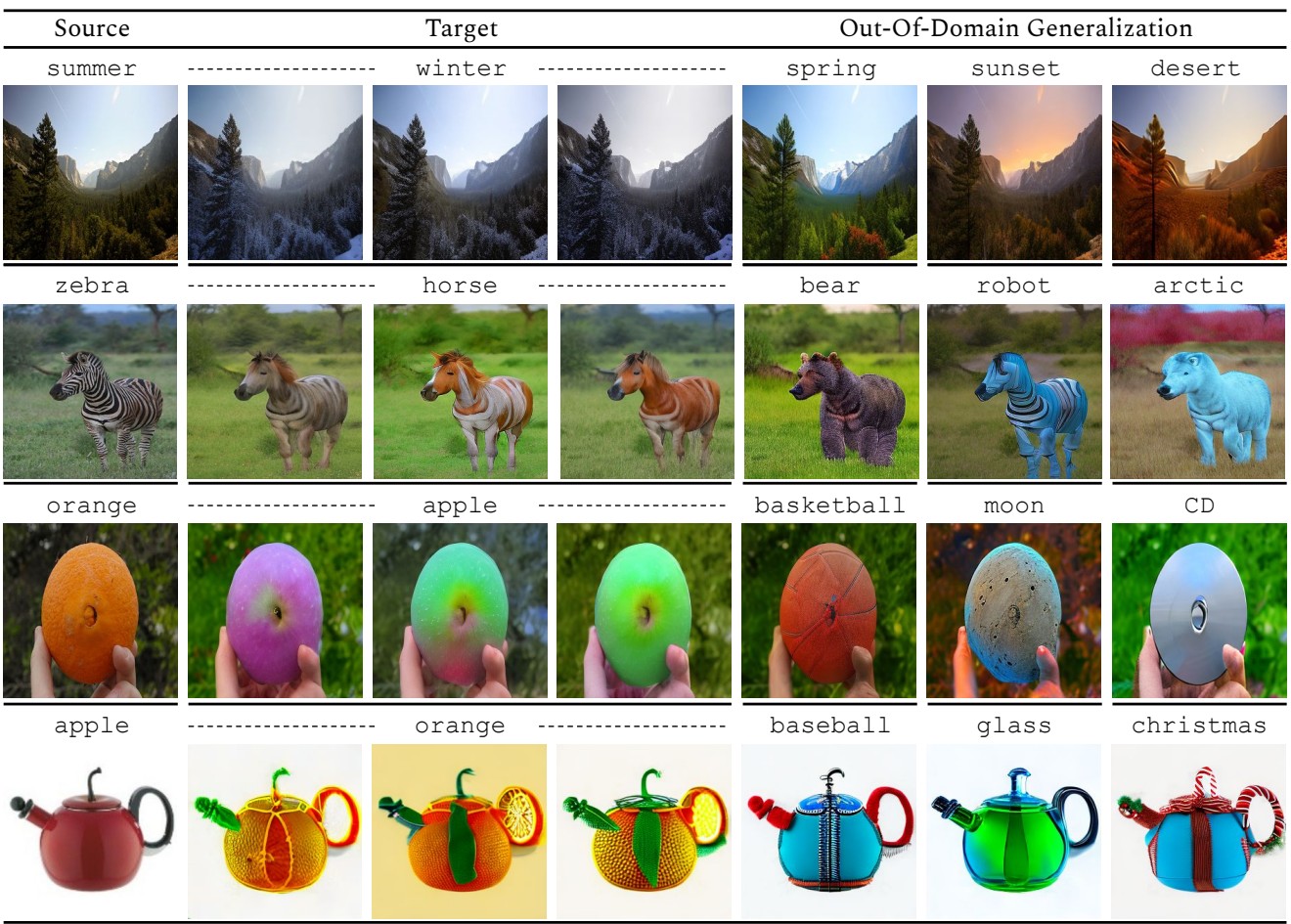

Figure 12: Additional examples of output diversity in the target domains and zero-shot generalization to out-of-domain distributions.

