# OpenReview forum: "CycleNet: Rethinking Cycle Consistency in Text-Guided Diffusion for Image Manipulation"
_NeurIPS.cc/2023/Conference — NeurIPS 2023 poster_

### Official Review · Reviewer_sYAZ · 2023-07-03

**Soundness:** 4 excellent
**Presentation:** 4 excellent
**Contribution:** 2 fair
**Rating:** 6
**Confidence:** 4

**Summary:**

The paper proposes CycleNet, a method for unpaired image-to-image translation using pretrained diffusion models. The main idea is to reconstruct the conditional images through a reverse process. The idea seems reasonable, and the paper is well-organized, with sufficient experimental results.

**Strengths:**

The idea seems reasonable, and the paper is well-organized, with sufficient experimental results.

**Weaknesses:**

1. The concept of cycle construction has been extensively studied in the field, and this work follows a similar idea, which weakens its contribution.
2. In CycleGAN, the reconstruction constraint $L_{x \rightarrow y  \rightarrow x}$ plays a vital role. However, in FastCycleNet, this constraint is omitted, yet the performance does not show significant drops. It would be helpful if the authors could provide an explanation for this observation.
3. For some variable notation (such as ($\bar{x}_o$), authors had better provide their definitions, It would be beneficial for readers to quickly understand these notations.

**Questions:**

Please see in weakness

**Limitations:**

Yes

---

> ### Author Rebuttal · Authors · 2023-08-10
>
> We are happy that the reviewer found our paper "well-organized, with sufficient experimental results.". We appreciate their constructive feedback on our efforts.
>
> ### Question 1
> > Q1: The concept of cycle construction has been extensively studied in the field, and this work follows a similar idea, which weakens its contribution.
>
> **Response to Q1**: We notice that there is an inconsistent view of our novelty across reviewers. We believe this might be addressed with more clarity and better comparisons to related efforts. We kindly redirect the reviewer to **General response 2** in the General Response for our clarifications. In short, we hope to bring it to reviewers’ attention that CycleNet presents the first effort that guarantees cycle consistency in **unpaired** image-to-image translation using **pre-trained diffusion models**, which is theoretically motivated and empirically promising.
>
> ### Question 2
> > Q2: In CycleGAN, the reconstruction constraint Lx-y-x plays a vital role. However, in FastCycleNet, this constraint is omitted, yet the performance does not show significant drops. It would be helpful if the authors could provide an explanation for this observation.
>
> **Response to Q2**: We believe that the reconstruction loss $L_{x-x}$ can still help the model to maintain a certain level of consistency. The Stable Diffusion is originally trained on reconstruction with text conditions but without image conditions, losing its consistency when image conditions are introduced. In FastCycleNet, we explicitly train the diffusion model to reconstruct an image given on the image condition. Therefore, the performance does not show significant drops, while there is a noticeable gap according to our ablation study.
>
> ### Question 3
> > Q3: For some variable notation (such as $\bar{x_0}$), authors had better provide their definitions, It would be beneficial for readers to quickly understand these notations.
>
> **Response to Q3**: $\bar{x_0}$ refers to the reconstructed input image $x_0$ from back translation. We thank the reviewer for the suggestions! We will make sure to provide more concrete definitions for the notations we introduced.

---

### Official Review · Reviewer_yCsT · 2023-07-06

**Soundness:** 2 fair
**Presentation:** 2 fair
**Contribution:** 2 fair
**Rating:** 2
**Confidence:** 5

**Summary:**

The paper introduces CycleNet, a new method that enhances image manipulation by incorporating cycle consistency into diffusion models. The paper addresses the challenge of unpaired image-to-image translation and aims to provide a consistent and intuitive interface for this task.

**Strengths:**

1.Originality: The paper introduces CycleNet, a new method that incorporates cycle consistency into diffusion models for image manipulation. This approach combines the concept of cycle consistency with text-guided diffusion models.

2. Quality: The paper provides a comprehensive evaluation of CycleNet's performance using various quantitative metrics such as FID, CLIP Score, LPIPS, PSNR, SSIM, and L2.  The results also support the claims of the paper.

3. Clarity: The paper is well-written and presents the concepts, methodology, and findings.

**Weaknesses:**

1.Novelty is limited. The concept of combing cycle-consistency constraint and diffusion model for image-to-image translation has been widely explored by other works, such as UNIT-DDPM, Dual Diffusion Implicit Bridges (DDIBs), cyclediffusion and so on.

2.Lack of Comparative Analysis between the proposed method with other existing cycle-consistency based diffusion models. The part of 6.3 is not sufficient to illustrate the key difference and novelty of the proposed method.

3.The results are low-quality and far from satisfaction. For the classic image-to-image translation tasks, i.e., winter-to-summer and horse-to-zebra, I do not find any improvement over existing ui2i methods. In addition, the results of CUT aand CycleGAN are questionable, especially for winter-to-summer task, they are too bad compared with the results from their original papers.

4.The results of coffee-to-empty also do not convince me as it seems to only transfer the rgb image to gray image. Similar results also happen in the horse-to-zebra.

**Questions:**

1. Why there lack the results of strong baseline CycleGAN in Table 1?

2. Could the authors consider conducting a user study to evaluate the subjective quality and usability of CycleNet? This would provide insights into how users perceive the generated images and the overall user experience, further validating the practicality and effectiveness of CycleNet.

3. Could the authors provide more information about the computational efficiency of CycleNet? Specifically, details about the training and inference times, as well as the hardware specifications used, would be helpful in understanding the practicality of CycleNet for real-world applications.

**Limitations:**

1. Novelty is limited.

2. Poor results are far from satisfaction as the strong generation ability of the diffusion model.

---

> ### Author Rebuttal · Authors · 2023-08-10
>
> We are happy that the reviewer finds our paper “well-written”, and that “the results also support the claims of the paper”. We are more than grateful for the reviewer’s feedback, many of which has been integrated into our paper! We sincerely hope there will be engaged communication during our discussion, and that with our clarifications and updated results, the reviewer would consider re-evaluating our work!
>
> ### Weakness 1 and 2
> > W1: Novelty is limited, and Lack of Comparative Analysis.
>
> **Response to W1 and W2**: We notice that there is an inconsistent view of our novelty across reviewers. We believe this might be addressed with more clarity and better comparisons to related efforts. We kindly redirect the reviewer to **General response 2** in the General Response for our clarifications. In short, we hope to bring it to reviewers’ attention that CycleNet presents the first effort that **explicitly enforces cycle consistency constraint** in unpaired image-to-image translation using pre-trained diffusion models.
>
> DDIB is the theoretic foundation for various diffusion-based image translation models, and CycleDiffusion is an extension of the vanilla DDIB [46]. We thus compared with CycleDiffusion in our main paper. Unfortunately, UNIT-DDPM does not have any public implementation, and can not be reproduced given the limited information in the paper.
>
> ### Weakness 3
>
> > W3.1: The results are low-quality and far from satisfaction. For the classic image-to-image translation tasks, i.e., winter-to-summer and horse-to-zebra, I do not find any improvement over existing ui2i methods.
>
> **Response to W3.1**: We provide additional experiments on high-resolution examples in Appendix. Our observation is that CycleNet performs well, especially on complex scenes with multiple objects.
>
> > W3.2: In addition, the results of CUT aand CycleGAN are questionable, especially for winter-to-summer task, they are too bad compared with the results from their original papers. The results of coffee-to-empty also do not convince me as it seems to only transfer the rgb image to gray image. Similar results also happen in the horse-to-zebra.
>
> **Response to W3.2**: We use exactly the same images presented in the original papers without modifications. The quantitative results for horse-to-zebra are generated using the official model weights released by the authors.
>
> ### Question 1
> > Q1: Why there lack the results of strong baseline CycleGAN in Table 1?
>
> **Response to Q1**: We thank the reviewer for reminding us of this missing baseline. The qualitative comparisons to CycleGAN are already provided in the main paper. We apologize for missing this baseline in quantitative comparison and hereby provide the experimental results as requested.
>
> - For Summer-to-Winter:
>
> |             | **FID** | **FID CLIP** | **CLIP Sim** | **LPIPS** | **PSNR** | **SSIM** | **L2** |
> |-----------|-----------|-------------------|-------------------|---------------|---------------|--------------|----------|
> | CycleGAN | 133.16 (+53.15) | 18.85 (+4.71)  | 22.07 (-3.05) | 0.2 (+0.07)  | 16.27 (-6.18) | 0.39 (-0.18) | 3.62 (+2.72) |
>
> - For Horse-to-Zebra:
>
> |             | **FID** | **FID CLIP** | **CLIP Sim** | **LPIPS** | **PSNR** | **SSIM** | **L2** |
> |-----------|-----------|-------------------|-------------------|---------------|---------------|--------------|----------|
> | CycleGAN | 77.18 (-3.57) | 27.69 (+1.66)  | 28.07 (-0.84) | 0.25 (+0.02)  | 18.53 (-2.3) | 0.67 (+0.15) | 1.39 (+0.25) |
>
> In the quantitative evaluation, we observe that CycleNet outperforms CycleGAN in most of the metrics.
>
> ### Question 2
> > Q2: Could the authors consider conducting a user study to evaluate the subjective quality and usability of CycleNet? This would provide insights into how users perceive the generated images and the overall user experience, further validating the practicality and effectiveness of CycleNet.
>
> **Response to Q2**: We thank the reviewer for this suggestion and hereby provide the human evaluation results as requested. **Our human study has been approved by the Institution’s Institutional Review Board (IRB)**, who considered this study exempt from ongoing review.
>
> - **Setup**: We recruited 12 participants to vote for 30 sets of generated images (15 from summer to winter and 15 from horse to summer).
> > In this task, you'll assess an AI model's performance in translating a summer Yosemite photo into a winter version. You'll review the provided summer input photo and a set of images generated by different AI models. Your evaluation will focus on two aspects: Translation Quality: Assess the image's translation quality, considering its visual quality, depiction of a winter scene, and adherence to common expectations. Consistency is not a priority here, so changes in structure or missing objects can be overlooked. Translation Consistency: Evaluate the image's consistency in translation. Check if it faithfully maintains similarity to the original photo in terms of object count, types, patterns, and textures. The translation quality can be overlooked in this evaluation. You will not be limited to choosing only 1 out of all images, but please restrict your number of choices to under 3 for each set.
> - **Results**: Our human study results show that our model is preferred by human users, both in terms of consistency and quality. Due to the limited space, we will update the tabled results in the next iteration.
>
> ### Question 3
>
> > Q3: Could the authors provide more information about the computational efficiency of CycleNet? Specifically, details about the training and inference times, as well as the hardware specifications used, would be helpful in understanding the practicality of CycleNet for real-world applications.
>
> **Response to Q3**: The resulting algorithm is also practical in applications, relying on very little data (~2K) with minimal computational requirements (1GPU). We refer to Appendix B for more details.

---

> ### Comment · Reviewer_yCsT · 2023-08-19
> **Final comments after reading authors' response**
>
> My final rating will still be Strong Reject. The reasons mainly include three aspects: 1. limited novelty as the cycle-consistency has been widely explored in existing literatures; 2. poor image results as image editing quality does not improve with cycle-consistency and numerous edits appear unreasonable.; 3. the comparison results of existing methods are questionable.

---

> > ### Author Response · Authors · 2023-08-19
> > **Response to Reviewer yCsT's Comments**
> >
> > We thank the reviewer for their prompt response to us! We would like to confirm a few factual understandings so there is no miscommunication over the review process.
> >
> > ### Questionable Results
> > **Could the reviewer confirm that you have checked (1) our updated results in the General response and (2) the original CUT paper** (https://arxiv.org/abs/2007.15651)? Our presented images for CUT and CycleGAN in Figure 2 are **exactly the original copy** of the original paper (Horse->Zebra: Fig 3 & Summer->Winter: Fig 11). While the reviewer commented that “they are too bad compared with the results from their original papers”, we did use the same copy.
> >
> > ### Novelty
> >
> > **Could the reviewer confirm that they have read our discussion in General Response 2 (expanded below)?** If so, we would love to receive clear pointers to additional research that we missed in the discussion. We hope to know if the reviewer agrees that CycleNet presents the first effort that explicitly enforces cycle consistency constraints in image editing using pre-trained diffusion models.
> >
> > **Q: Has cycle-consistency constraint been explicitly integrated into diffusion models?**
> >
> > Not yet, to the best of our knowledge.
> >
> > Reviewer yCsT mentioned that:
> > > “The concept of **combing cycle-consistency constraint** and **diffusion model** for image-to-image translation has been widely explored by other works”
> >
> > We believe there is a factual misunderstanding. To the best of our knowledge, this is the first effort that **explicitly integrates cycle consistency constraint** in unpaired image-to-image translation using pre-trained diffusion models.
> >
> > We expand on what we have discussed in Section 6.3 regarding the several existing efforts to study cycle consistency in diffusion-based I2I translation.
> > - UNIT-DDPM ([39], Apr 2021) made an initial attempt in the unpaired I2I setting, training two DPMs and two translation functions from scratch. Cycle consistency losses are introduced in the translation functions during training to regularize the reverse processes. At inference time, the image generation does not depend on the translation functions, but only on the two DPMs in an iterative manner, leading to sub-optimal performance. This preprint does not have any public implementation, and can not be reproduced given the limited information in the paper.
> > - DDIB (Mar 2022): The key proposition states that exact cycle consistency is possible **assuming zero discretization error**. DDIB itself does not enforce any cycle consistency constraint:
> > > “DDIBs first obtain latent encodings for source images with the source diffusion model, and then decode such encodings using the target model to construct target images.”
> >
> > DDIB proposed one of the first theoretic analyses on the possibility to perform unpaired I2I translation with diffusion models and is the theoretic foundation for various diffusion-based image translation models, including prompt2prompt, NullText, and CycleDiffusion.
> >
> > - CycleDiffusion ([46], Nov 2022) proposes a zero-shot approach for image translation based on DDIB’s observation that a certain level of consistency could emerge from DMs. There is again no explicit treatment to encourage cycle consistency.
> > > “CycleDiffusion can be seen as an extension of DDIB approach to stochastic DPM” [46]
> > > “The method is based on the observation that cycle consistency naturally emerges from diffusion models, i.e., the same random noise leads to similar images. Therefore, there is no training objective that explicitly encourages cycle consistency.” (https://github.com/ChenWu98/cycle-diffusion/issues/16)
> >
> > If there is any similar prior work the reviewers are aware of, please let us know.
> >
> > **Q: Since a certain level of cycle consistency could emerge in diffusion models, do we benefit from enforcing it as a constraint?**
> >
> > Yes.
> >
> > We echo the comments of reviewer rhuF:
> > > “When we apply pre-trained DM models on I2I tasks, it is difficult or unpredictable to control the mask and attention maps to achieve the desired results.”
> >
> > While exact cycle consistency is possible assuming zero discretization error, the assumption generally doesn’t hold true. Pixel-level consistency remains to be challenging when there exists image conditioning, as can be shown in the failure cases of baselines presented in our paper, and the constantly changing backgrounds in image and video synthesis tasks.
> >
> > **Q: Why is it non-trivial to enforce cycle-consistency constraints in diffusion models?**
> >
> > Unlike GAN-based models, in which the translation between images is end-to-end, diffusion-based methods are iterative and lack such a simple and intuitive translation cycle. Also, for text-guided diffusion models, it is difficult to design a discriminator, which is a key component to ensure cycle consistency in GAN-based models. To address this limitation, we carefully formulated the problem in Section 2 and theoretically derived the training objectives in Section 3.

---

### Official Review · Reviewer_aEiE · 2023-07-06

**Soundness:** 3 good
**Presentation:** 3 good
**Contribution:** 3 good
**Rating:** 6
**Confidence:** 4

**Summary:**

This work aims at addressing the task of unpaired I2I translation with pre-trained diffusion models. Inspired by CycleGAN, authors incorporate cycle consistency into diffusion model to regularize process of image translation and proposed CycleNet. In addition it also contributes a multi-domain I2I translation dataset with object state change. Extensive experiments demonstrate that the proposed method can produce high-quality translation results with good consistency and is able to generalize to out-of-domain generation tasks by changing text prompts.

**Strengths:**

1. It is interesting and reasonable to introduce cycle consistency into diffusion models for training unpaired I2I translation models.
2. A new multi-domain I2I translation dataset for object state change with text prompt is collected.
3. Extensive experiments demonstrate the effectiveness of the proposed method.
4. Capbility to generalize to out-of-domain data by simply changing text prompt.

**Weaknesses:**

1. I wonder how CycleNet compares with CycleGAN. More comparison between CylceNet and CycleGAN should be provided. If their performance is similar, then why use a diffusion-based method instead of GAN-based for unpaired I2I translation task?
2. Authors are suggested to experiment with more diffusion models such as T2I Adapter to show its generalization on models.
3. Any explanation on why FastCycleNet perform better in several metrics? The coffee-to-empty translation result does not seem good, not only changing the color but also failing to become empty.

**Questions:**

See weakness.

**Limitations:**

Authors have clearly showed failure cases of the proposed method in the main text. But they do not address potential negative socital impact of this work.

---

> ### Author Rebuttal · Authors · 2023-08-10
>
> We are happy that the reviewer found it “interesting and reasonable to introduce cycle consistency into diffusion models”. We thank the reviewer for recognizing our novelty, our “extensive experiment”, and our contributions (out-of-domain generalization, new multi-domain dataset). We appreciate their constructive feedback on our efforts.
>
> ### Question 1
> > Q1.1: I wonder how CycleNet compares with CycleGAN. More comparison between CylceNet and CycleGAN should be provided.
>
> **Response to Q1.1**: We thank the reviewer for reminding us of this missing baseline. The qualitative comparisons to CycleGAN are already provided in the main paper. We apologize for missing this baseline in quantitative comparison and hereby provide the experimental results as requested.
>
> - For Summer-to-Winter:
>
> |             | **FID** | **FID CLIP** | **CLIP Sim** | **LPIPS** | **PSNR** | **SSIM** | **L2** |
> |-----------|-----------|-------------------|-------------------|---------------|---------------|--------------|----------|
> | CycleGAN | 133.16 (+53.15) | 18.85 (+4.71)  | 22.07 (-3.05) | 0.2 (+0.07)  | 16.27 (-6.18) | 0.39 (-0.18) | 3.62 (+2.72) |
>
> - For Horse-to-Zebra:
>
> |             | **FID** | **FID CLIP** | **CLIP Sim** | **LPIPS** | **PSNR** | **SSIM** | **L2** |
> |-----------|-----------|-------------------|-------------------|---------------|---------------|--------------|----------|
> | CycleGAN | 77.18 (-3.57) | 27.69 (+1.66)  | 28.07 (-0.84) | 0.25 (+0.02)  | 18.53 (-2.3) | 0.67 (+0.15) | 1.39 (+0.25) |
>
> In the quantitative evaluation, we observe that CycleNet outperforms CycleGAN in most of the metrics.
>
> > Q1.2: If their performance is similar, then why use a diffusion-based method instead of GAN-based for unpaired I2I translation task?
>
> **Response to Q1.2**: Diffusion-based methods are superior in terms of image quality, enable multi-resolution generation, and are more scalable toward large multimodal datasets. Additionally, our method also enables strong generalization to unseen domains, where CycleGAN can hardly perform. We also show in Figure 3 and 10 (Appendix) that the GAN-based methods are more prone to assigning zebra patterns to non-horse objects, indicating their limited understanding of the entities in the scene.
>
> ### Question 2:
> > Q2: Authors are suggested to experiment with more diffusion models such as T2I Adapter to show its generalization on models.
>
> **Response to Q2**: We were a bit lost by what the reviewer meant by “more diffusion models such as T2I Adapter”, so we hope to make sure to address the reviewer’s concern from the aspects of **diffusion models** or **diffusion model adapters**.
> - **More diffusion models**: To our knowledge, T2I-Adapter [30] is a plug-in-and-play adapter on pre-trained diffusion models (DMs) like stable diffusion (SD), rather than a diffusion model itself. If the reviewer is looking for additional experiments on more latent diffusion backbones, we would love to kindly point to Section E.1 in the Appendix, where we compare CycleNet on different versions of SD backbone, and confirm that our framework can generalize across different DM backbone.
> - **More diffusion models adapters**: We agree that our theoretical framework could potentially generalize to different adapters, not restricted to ControlNet. We would love to present additional experiments to support this view, but unfortunately, the training configurations are not available yet in the T2I-Adapter official GitHub, leading to a reproducibility issue. Also, T2I-Adapter does not take uncondition prompts, and we need non-trivial work to adapt it to our framework. Given the current status, we are unable to present quantitative experiments during the discussion period. We will be happy to update our paper once the training config of T2I-Adapter is publicly available to the community.
>
> ### Question 3
> > Q3.1: Any explanation on why FastCycleNet perform better in several metrics?
>
> **Response to Q3.1**: Great question! A quick explanation: There is a known trade-off between translation quality and translation consistency [52]. As FastCycleNet does not enforce the cycle consistency loss $L_{x-y-x}$, it becomes better at image quality and translation quality, compared to the full CycleNet that enforces this loss.
>
> We want to kindly refer the reviewer to Section 5.3 in the main paper, where we presented an ablation study. In this section, FastCycleNet is identical to the Invariance-Only version of CycleNet, which achieves better translation performance, as there is less trade-off between translation and consistency when the cycle consistency loss is removed. On the other hand, FastCycleNet suffers from relatively lower performance in terms of consistency metrics.
>
> > Q3.2 The coffee-to-empty translation result does not seem good, not only changing the color but also failing to become empty.
>
> **Response to Q3.1**: We hope to address the reviewer’s concern in the following aspects:
> - **Addressing the problem**: We kindly redirect the reviewer to **General response 1** in the General Response for our solution and updated results. In short, we address this with a simple but effective solution at inference. Our updated experimental results show that this issue of **hue shifting can be addressed with a minimal trade-off in image quality and translation quality**. We sincerely hope that the reviewer can re-evaluate our work.
> - **Emphasizing our dataset contribution**: We would like to emphasize our contribution to identify a challenging problem in image translation with object state changes, and to provide the first possible dataset to study this problem. We have observed that the task has posed a formidable challenge for all the models we tested. While our model performs satisfactorily in “Empty-to-Full” tasks, we acknowledge that the “Full-to-Empty” task remains challenging for our current solution. We hope the community will allocate more attention to this problem of image translation with object state changes.

---

> > ### Comment · Reviewer_aEiE · 2023-08-22
> > **Response to rebuttal**
> >
> > I appreciate the efforts made by authors in addressing most of my concerns with more experimental results and explanation. So I would like to keep my original rating.

---

> > > ### Author Response · Authors · 2023-08-22
> > > **Response to Reviewer aEiE's Comments**
> > >
> > > We thank the reviewer for the positive feedback! We are pleased that we have addressed your concerns regarding the potential experiment settings and baselines, and that the reviewer found our explanations convincing. The updates will be reflected in the next iterations. Your suggestions are invaluable in enhancing our work.

---

### Official Review · Reviewer_Tt5a · 2023-07-07

**Soundness:** 3 good
**Presentation:** 2 fair
**Contribution:** 2 fair
**Rating:** 5
**Confidence:** 4

**Summary:**

This paper presents a method that tackles the challenge of consistent image-to-image (I2I) translation using diffusion models (DMs) and unpaired image conditioning and text prompts. The proposed approach incorporates Cycle Consistency Regularization to ensure cycle consistency and Self-Regularization to generate images that align with the target domain. Furthermore, the paper introduces a multi-domain I2I translation dataset that encompasses object state changes.

**Strengths:**

a)	From the visual results, the proposed CycleNet can well preserving the structures of the input image.

b)	The author introduces a multi-domain I2I translation dataset for object state changes.

**Weaknesses:**

a)	The experimental results are quite strange.

i.	In Figure 2, Figure 3, Figure 5, and many images in the supplementary materials, The results of CycleNet show severe hue shifting. This is a serious issue, and I suggest the author analyze and address this problem.

b)	The author's comparison methods are not comprehensive. Many editing methods based on diffusion have been proposed, such as pix2pix-zero, masactrl, etc. I suggest the author compare these methods.

c)	Although CycleNet aims to handle unpaired I2I translation tasks, it still requires paired datasets for training. So, how well does CycleNet generalize to unseen domains (such as variations of other unseen species, etc.)?

**Questions:**

Please refer to the weaknesses part.

**Limitations:**

Please refer to the weaknesses part.

---

> ### Author Rebuttal · Authors · 2023-08-10
>
> We thank the reviewer for recognizing our dataset contributions and our model’s capability to preserve the structures of input images. We are happy that much of the constructive feedback has been integrated into our paper!
>
> ### Question 1
> > Q1: The experimental results are quite strange. In Figure 2, Figure 3, Figure 5, and many images in the supplementary materials, The results of CycleNet show severe hue shifting. This is a serious issue, and I suggest the author analyze and address this problem.
>
> **Response to Q1**: We thank the reviewer for pointing out this hue-shifting issue, and kindly redirect the reviewer to **General response 1** in the General Response for our solution and updated results. In short, we address this with a simple but effective solution at inference time, and the updated results are as follows.
>
> Previously, when we are sampling the image at inference time, we start from a **random noise** $\epsilon$. In our updated method, we initialize the process with the **latent noised input image** $z_t$, collected using equation (1). This is in fact a commonly used inference time trick, which is also adopted in some of the baselines [28, 46] to improve consistency in image translation.
>
> Our updated experimental results show that this issue of **hue shifting can be addressed with minimal trade-off in image quality and translation quality**. We sincerely hope that the reviewer can re-evaluate our work.
>
> ### Question 2
> > Q2: The author's comparison methods are not comprehensive. Many editing methods based on diffusion have been proposed, such as pix2pix-zero, masactrl, etc. I suggest the author compare these methods.
>
> **Response to Q2**: We thank the reviewer for pointing out these concurrent work to us. We will for sure give a detailed discussion and evaluation in the next iteration. We hereby provide the experimental results as requested, quantitatively in the table below and qualitatively attached in the PDF.
>
> - For Summer-to-Winter:
>
> |                    | **FID** | **FID CLIP** | **CLIP Sim** | **LPIPS ** | **PSNR** | **SSIM** | **L2** |
> |-----------------|-----------|-------------------|-------------------|---------------|---------------|--------------|----------|
> | pix2pix-zero| 311.03 (+231.02) | 81.54 (+67.4)  | 22.03 (-3.09) | 0.57 (+0.44)  | 14.31 (-8.14) | 0.32 (-0.24) | 5.08 (+4.18) |
> |    masactrl   | 106.91 (+26.9) | 52.38 (+38.24)  | 20.79 (-4.33) | 0.36 (+0.23)  | 16.22 (-6.23) | 0.36 (-0.21) | 3.71 (+2.81) |
>
> - For Horse-to-Zebra:
>
> |                    | **FID** | **FID CLIP** | **CLIP Sim** | **LPIPS ** | **PSNR** | **SSIM** | **L2** |
> |-----------------|-----------|-------------------|-------------------|---------------|---------------|--------------|----------|
> | pix2pix-zero| 377.44 (+296.69) | 86.21 (+60.18)  | 24.37 (-4.54) | 0.67 (+0.4)  | 11.18 (-9.65) | 0.19 (-0.33) | 3.85 (+2.71) |
> |    masactrl   | 333.17 (+252.42) | 68.31 (+42.28)  | 21.15 (-7.76) | 0.40 (+0.13)  | 16.31 (-4.52) | 0.37 (-0.15) | 1.83 (+0.69) |
>
> As can be seen from the results, CycleNet is able to outperform these newly proposed baselines by a large margin over all metrics and dimensions.
>
> ### Question 3
> > Q3: Although CycleNet aims to handle unpaired I2I translation tasks, it still requires paired datasets for training. So, how well does CycleNet generalize to unseen domains (such as variations of other unseen species, etc.)?
>
> **Response to Q3**: We are afraid that there might be a factual misunderstanding. We hope the following paragraphs can ensure there is no misunderstanding regarding the **setting** or **performance** on the generalizability of CycleNet to unseen domains.
>
> - **Regarding the setting**: To the best of our understanding, paired datasets typically refer to datasets containing image pairs at the **instance-level**, representing images before and after translation [26, 53]. We first kindly emphasize that CycleNet does **NOT** require such paired datasets for training, while ControlNet does.
> - **Regarding the performance**: However, if the reviewer's remark pertains to **domain-level** pairs, we would also love to kindly remind the reviewer of our experiments of the CycleNet’s generalization ability in Section 5.2 in the main paper. Figure 5 shows that CycleNet can leverage the pre-trained diffusion backbone, and generalize well to unseen domains with a simple change of the textual prompt.

---

> > ### Comment · Reviewer_Tt5a · 2023-08-20
> > **Official Comment by Reviewer Tt5a**
> >
> > I would like to convey my appreciation for the authors' dedicated efforts in responding to my inquiries. In response to the Hue-Shifting issue, the authors proposed a solution to mitigate this problem. From the attached PDF provided by the authors, it can be seen that the hue shifting has indeed been alleviated to some extent. In addition, the authors have also included experimental results for pix2pix-zero and masactrl. Therefore, I will raise my score to 5.

---

> > > ### Author Response · Authors · 2023-08-20
> > > **Response to Reviewer Tt5a's Comments**
> > >
> > > We thank the reviewer for your positive feedback and prompt response! We are pleased that we have addressed your concerns regarding the hue-shifting issues and missing baselines you mentioned. These updates will be certainly reflected in the next version. Your suggestions are invaluable in enhancing our work.

---

### Official Review · Reviewer_rhuF · 2023-07-10

**Soundness:** 3 good
**Presentation:** 3 good
**Contribution:** 3 good
**Rating:** 5
**Confidence:** 4

**Summary:**

This paper introduces cycle consistency in diffusion models to achieve regularization in image manipulation. It allows out of domain image generation with text prompt modification, and can be trained with very little data (~2K) with minimal computational requirements (1GPU). When we apply pre-trained DM models on I2I tasks, it is difficult or unpredictable to control the mask and attention maps to achieve the desired results. Cyclic consistency regularization allows us to address this. Experiments are conducted on many datasets such as Yosemiti summer-winter, horse-zebra, etc. On standard metrics such as CLIP, FID, and LPIPS, the proposed method outperforms other baselines such as CycleGAN, TEX2LIVE, etc.

**Strengths:**

1) There has not been much regularization in diffusion models except for guidance driven by the conditional text. It is nice to see the celebrated Cycle-Consistency loss developed for diffusion models.

2) The paper clearly motivates and states the problem. It is nice to see that the proposed method nicely inherits the controlnet architecture to incorporate the cycle-consistency loss function during training.


**Weaknesses:**

1) While enforcing cyclic consistency we are feeding the actual input image as the conditional input. This seems a bit strong and it appears that the diffusion model may be overfitting to the conditional input.

2) It is not clear how one obtains the negative prompt c^{-1}. The method takes a prompt c_{text}=(c^{+}, c^{-}), where c^{+} is the prompt that drives the diffusion process towards the images that are associated with it, and the negative prompt c^{-} drives the process away from these images. The paper only mentions this once without sufficient details on how this is done.

3) The proposed method heavily relies on controlNet and uses the original image to impose the Cyclic-Consistency loss. It may be worth just testing with standard ControlNet with canny edge-map as conditional input and appropriate text prompt.



**Questions:**

Overall, the paper looks at an important problem and brings cyclic consistency to diffusion models. Please see my concerns above.

**Limitations:**

No concerns.

---

> ### Author Rebuttal · Authors · 2023-08-09
>
> We thank the reviewer for recognizing our novelty (“there has not been much regularization in diffusion models”) and contributions (out-of-domain generalization, minimal data, and compute). We appreciate their constructive feedback on our efforts.
>
> ### Question 1
> > Q1: While enforcing cyclic consistency we are feeding the actual input image as the conditional input. This seems a bit strong and it appears that the diffusion model may be overfitting to the conditional input.
>
> **Response to Q1**: We first note that our main focus and contribution is to enable consistency in text-guided manipulation with pre-trained diffusion models (DMs). While pre-trained DMs are widely known for their unprecedented capability in high-quality text-to-image synthesis, pixel-level consistency remains to be challenging when there exists image conditioning, as can be shown in the constantly changing backgrounds in image and video synthesis tasks.
>
> We acknowledge this trade-off between **translation consistency** and **translation quality** in image translation [52], as has been discussed in Section 5.4 (Limitations). While we are aware of the current limitation, we firmly recognize the significance of addressing this matter in future research endeavors. In this context, methodologies such as employing local discriminators [54] hold great promise as inspiring solutions to tackle this concern in the future, which lies beyond the scope of this work.
>
> ### Question 2
> > Q2: It is not clear how one obtains the negative prompt c^{-1}. The method takes a prompt c_{text}=(c^{+}, c^{-}), where c^{+} is the prompt that drives the diffusion process towards the images that are associated with it, and the negative prompt c^{-} drives the process away from these images. The paper only mentions this once without sufficient details on how this is done.
>
> **Response to Q2**: Thank you for raising this clarity issue! We hope the following paragraphs can ensure there is no misunderstanding regarding the **source**, **implementation** and **mechanisms** of conditional (positive) and unconditional (negative) prompts.
>
> - **Regarding the source**: We kindly point to lines 151-155 where we described how we obtain conditional and unconditional prompts. In the context of unpaired I2I translation, the conditional prompt is a piece of textual description of the target domain, and the unconditioned prompt is that of the source domain. For example, to translate a summer image to winter, we can simply use a conditional prompt “summer” and an unconditional prompt “winter”. These prompts are pre-specified during training (see lines 174-179) and are provided during inference time. We will be sure to be clearer about this.
> - **Regarding the implementation**: We encode these prompts using a CLIP encoder similar to prior work (lines 156-157).
> - **Regarding the mechanism**: Negative prompts are a common interface in pre-trained DMs to remove undesirable semantic features in the unconditional sampling. We point to this tutorial [A1] for illustrative explanations. In CycleNet, we encode the conditional prompt in the frozen SD backbone. The unconditional prompt and the image condition are encoded in the adapter, so that the adapter learns to attend to regions that need to be modified.
>
> [A1] How does negative prompt work? https://stable-diffusion-art.com/how-negative-prompt-work/
>
> ### Question 3
> > Q3: The proposed method heavily relies on controlNet and uses the original image to impose the Cyclic-Consistency loss. It may be worth just testing with standard ControlNet with canny edge-map as conditional input and appropriate text prompt.
>
> **Response to Q3**: We thank the reviewer for pointing out this valid baseline to consider. We hereby provide the experimental results as requested, quantitatively in the table below and qualitatively attached in the PDF.
>
> - For Summer-to-Winter:
>
> |             | **FID** | **FID CLIP** | **CLIP Sim** | **LPIPS** | **PSNR** | **SSIM** | **L2** |
> |-----------|-----------|-------------------|-------------------|---------------|---------------|--------------|----------|
> |ControlNet+Canny| 338.24 (+258.14) | 83.26 (+69.12)  | 21.77 (-3.35) | 0.59 (+0.46)  | 6.05 (-16.4) | 0.09 (-0.48) | 11.3 (+10.4) |
>
> - For Horse-to-Zebra:
>
> |             | **FID** | **FID CLIP** | **CLIP Sim** | **LPIPS** | **PSNR** | **SSIM** | **L2** |
> |-----------|-----------|-------------------|-------------------|---------------|---------------|--------------|----------|
> |ControlNet+Canny| 397.71 (+316.42) | 77.68 (+51.65)  | 23.88 (-4.29) | 0.61 (+0.34)  | 7.37 (-13.46) | 0.07 (-0.45) | 4.89 (+3.75) |
>
> As can be seen from the results, standard ControlNet plus canny edge-map does not guarantee faithful and qualitative translation. This method might suffer from a significant loss of semantic information in the canny edge-maps, which only present the structural information.

---

> > ### Comment · Reviewer_rhuF · 2023-08-21
> > **Acknowledging the rebuttal**
> >
> > I thank the authors for the rebuttal. It addresses most of my concerns including the ControlNet experiments. I would like to keep my positive rating.

---

> > > ### Author Response · Authors · 2023-08-21
> > > **Response to Reviewer rhuF's Comments**
> > >
> > > We thank the reviewer for your response! We are pleased that we have addressed your concerns and your evaluation remains positive. The updates will be reflected in the next version and your suggestions are invaluable for our work!

---

### Author Rebuttal · Authors · 2023-08-10

### General response 1: Hue-Shifting and Sub-Optimal Performance on ManiCups

We thank the reviewers for pointing out this hue-shifting issue. We address this with a simple but effective solution at inference time and the updated results are as follows.

**Method**: Previously, when we are sampling the image at inference time, we start from a **random noise** $\epsilon$. In our updated method, we initialize the process with the **latent noised input image** $z_t$, collected using equation (1). This is in fact a commonly used trick at the inference time, which is also adopted in some of the baselines [28, 46] to improve consistency in image translation.

**Results**: Following [46], we set $t=100$ to collect the noised latent for a fair comparison. We label these updated models with “(Init)” in the tables. We hereby provide updated experimental results, quantitatively in the table below and qualitatively in the attached PDF.

- For Summer-to-Winter and Horse-to-Zebra

|                    | **FID** | **FID CLIP** | **CLIP Sim** | **LPIPS** | **PSNR** | **SSIM** | **L2** |
|-----------------|-----------|-------------------|-------------------|---------------|---------------|--------------|----------|
| Summer-to-Winter, CycleNet (Init) | 82.52 (+0.73) | 17.54 (+3.4) | 23.32 (-1.32) | 0.13 (-0.07)  | 22.42 (+4.07) | 0.57 (+0.07) | 0.9 (-0.52) |
| Summer-to-Winter, FastCycleNet (Init) | 82.48 (+2.47) | 17.61 (+3.38) | 23.62 (-1.5) | 0.14 (-0.1) | 22.45 (+5.24) | 0.57 (+0.11) | 0.91 (-0.53) |
| Horse-to-Zebra, CycleNet (Init) | 81.69 (-0.43) | 28.11 (-0.1) | 28.91 (+0.96) | 0.27 (-0.04) | 20.42 (+1.57) | 0.52 (+0.02) | 1.14 (-0.09) |
| Horse-to-Zebra, FastCycleNet (Init) | 80.75 (-9.71) | 27.23 (+1.2) | 27.36 (-0.81) | 0.32 (0) | 19.29 (-1.54) |  0.51 (+0.05) | 1.31 (-0.02) |

- For CycleNet (Init) on Coffee-to-Empty and Empty-to-Juice:

|                    | **FID** | **FID CLIP** | **CLIP Sim** | **LPIPS** | **PSNR** | **SSIM** | **L2** |
|-----------------|-----------|-------------------|-------------------|---------------|---------------|--------------|----------|
| Coffee-to-Empty |  95.24 (-27.68) | 28.79 (0) | 27.54 (+0.88) | 0.14 (-0.04) | 21.85 (-0.32) | 0.78 (+0.08) | 1.92 (-0.09) |
| Empty-to-Juice | 79.02 (-47.83) | 23.42 (-9.31) | 27.75 (+1.29) | 0.17 (-0.04) | 20.18 (-1.58) | 0.76 (+0.06) | 2.27 (+0.19) |

The qualitative examples (in PDF) have shown that this simple initialization trick can satisfyingly mediate the hue-shifting issue, resulting in improved performance, especially in ManiCups tasks. In the quantitative evaluation, except for a very marginal increase in FID and decrease in CLIPScore in some cases, we observe better performance in most of the metrics (especially in terms of ManiCups tasks and the consistency metrics), compared to previous non-initialized CycleNets.

Overall, we show that this issue of **hue shifting can be addressed with a minimal trade-off in image quality and translation quality**. We sincerely hope that the reviewer can re-evaluate our work.

### General response 2: Novelty, and Cycle Consistency in Diffusion Models

We notice that there are different views in terms of the novelty of this work across reviewers. Especially, Reviewer yCsT mentioned that:
> “The concept of **combing cycle-consistency constraint** and **diffusion model** for image-to-image translation has been widely explored by other works”

We believe there is a factual misunderstanding. To the best of our knowledge, this is the first effort that **explicitly integrates cycle consistency constraint** in unpaired image-to-image translation using pre-trained diffusion models.

**Q: Has cycle-consistency constraint been explicitly integrated into diffusion models?**
Not yet, to the best of our knowledge.

We expand on what we have discussed in Section 6.3 regarding the several existing efforts to study cycle consistency in diffusion-based I2I translation.
- UNIT-DDPM ([39], Apr 2021) made an initial attempt in the unpaired I2I setting with cycle consistency. At inference time, the image generation does not depend on the cycle-consistent translation functions, leading to sub-optimal performance. Also, this preprint does not have any public implementation, and can not be reproduced given the limited information in the paper.
- DDIB (Mar 2022): The key proposition states that exact cycle consistency is possible **assuming zero discretization error**. DDIB itself does not enforce any cycle consistency constraint:
> “DDIBs first obtain latent encodings for source images with the source diffusion model, and then decode such encodings using the target model to construct target images.”
- CycleDiffusion ([46], Nov 2022) proposes a zero-shot approach for image translation based on DDIB’s observation that a certain level of consistency could emerge from DMs. There is again no explicit treatment to encourage cycle consistency.
> “The method is based on the observation that cycle consistency naturally emerges from diffusion models, i.e., the same random noise leads to similar images. Therefore, there is no training objective that explicitly encourages cycle consistency.” (https://github.com/ChenWu98/cycle-diffusion/issues/16)

If there is any similar prior work the reviewers are aware of, please let us know.

**Q: Do we benefit from enforcing it as a constraint? Why is it non-trivial to do so in diffusion models?**

We echo the comments of reviewer rhuF:
> “When we apply pre-trained DM models on I2I tasks, it is difficult or unpredictable to control the mask and attention maps to achieve the desired results.”

While exact cycle consistency is theoretically possible, the assumption generally doesn’t hold true. Consistency remains to be challenging when there exists image conditioning. Unlike GAN-based models, diffusion-based methods are iterative and lack such a simple and intuitive translation cycle. Also, for text-guided diffusion models, it is difficult to design a discriminator.

---

### Comment · Area_Chair_FHyR · 2023-08-18
**Diverging scores**

Dear reviewers,

The paper got diverging scores. The authors provided their response.

Please engage into the discussion with authors, fellow reviewers and provide your final recommendation?

Reviewers Tt5a, yCsT are on the negative side. Do the other reviews and the authors' response change your mind?

Reviewers rhuF, aEiE, and sYAZ are on the positive side. Do the comments from Tt5a, yCsT influence your recommendation?

Please discuss.

Your AC

---

### Decision · Program_Chairs · 2023-09-21

**Decision:**

Accept (poster)

**Comment:**

The paper received borderline scores slightly leaning positive. All the reviewers acknowledged the key idea of the paper---regularize diffusion models with cycle consistency. The reviewers had concerns with the evaluation, in particular, reviewer Tt5a mentioned the hue-shifting problem visible in the results. In their response the authors addressed this and other concerns. The reviewer yCsT maintained a strongly negative score as they believe the literature contains a lot of related works in which similar ideas are explores. The authors disagreed with that statement. This concern also didn't sway the other reviewers to the negative side. Hence, while there is merit in yCsT's comments, the AC tends to agree with the positive reviewers and decides to recommend acceptance of the manuscript.